# Loss of CREBBP and KMT2D cooperate to accelerate lymphomagenesis and shape the lymphoma immune microenvironment

Jie Li [1,16], Christopher R. Chin [1,2,3,16], Hsia-Yuan Ying[1,16], Cem Meydan [2,3], Matthew R. Teater[1], Min Xia[1], Pedro Farinha [4], Katsuyoshi Takata[5], Chi-Shuen Chu[6], Yiyue Jiang [7,8], Jenna Eagles[7], Verena Passerini[9], Zhanyun Tang[6], Martin A. Rivas [1], Oliver Weigert [9], Trevor J. Pugh [7,8,10], Amy Chadburn[11], Christian Steidl [5], David W. Scott [12], Robert G. Roeder [6], Christopher E. Mason [2,3,13,14], Roberta Zappasodi [1,15], Wendy Béguelin [1] ✉ & Ari M. Melnick [1] ✉

Despite regulating overlapping gene enhancers and pathways, *CREBBP* and *KMT2D* mutations recurrently co-occur in germinal center (GC) B cell-derived lymphomas, suggesting potential oncogenic cooperation. Herein, we report that combined haploinsufficiency of *Crebbp* and *Kmt2d* induces a more severe mouse lymphoma phenotype (vs either allele alone) and unexpectedly confers an immune evasive microenvironment manifesting as CD8+ T-cell exhaustion and reduced infiltration. This is linked to profound repression of immune synapse genes that mediate crosstalk with T-cells, resulting in aberrant GC B cell fate decisions. From the epigenetic perspective, we observe interaction and mutually dependent binding and function of CREBBP and KMT2D on chromatin. Their combined deficiency preferentially impairs activation of immune synapse-responsive super-enhancers, pointing to a particular dependency for both co-activators at these specialized regulatory elements. Together, our data provide an example where chromatin modifier mutations cooperatively shape and induce an immune-evasive microenvironment to facilitate lymphomagenesis.

Epigenetic dysregulation plays a central role in malignant transformation and in encoding tumor phenotypes[1-3]. Chromatin modifier gene mutations are highly frequent in follicular lymphoma (FL) and diffuse large B-cell lymphoma (DLBCL), the two most common non-Hodgkin lymphomas (NHLs)[4-6]. Both of these tumors arise mostly from B cells transiting through the germinal center (GC) reaction during the humoral immune response[7].

The two most recurrently mutated genes in these tumors are *KMT2D* and *CREBBP*, which are believed to be founding events in lymphomagenesis[8,9]. KMT2D is a histone methyltransferase that selectively mediates H3K4 mono- and di-methylation (H3K4me1/

me2) primarily at enhancers to prime their activation[10,11]. KMT2D is most commonly affected by heterozygous nonsense mutations that yield a truncated protein lacking the enzymatic SET domain. CREBBP is a histone acetyltransferase that mediates enhancer activation through acetylation of H3K27 and other residues[12-14]. CREBBP loss of function in FL and DLBCL is caused by either missense mutations in the histone acetyltransferase domain (HAT) that inactivate enzymatic activity, or by truncating mutations. Homozygous conditional knockout of *Kmt2d* or *Crebbp* in mouse GC B cells leads to focal reduction of H3K4me1 and H3K27ac respectively, preferentially at gene enhancers[15-20]. Many of the affected genes overlap and are

involved in immune signaling pathways, which play important roles in GC physiology.

It is a generally accepted principle that within a given tumor, oncogenic hits that engage the same pathways are generally mutually exclusive to each other. One example of this are the mutually exclusive mutations in TET2 and IDH enzymes in acute myeloid leukemia[21]. The reason for their exclusivity is believed due to their both inducing gene promoter cytosine hypermethylation as an important part of their oncogenic functions. Likewise, mutations in *CREBBP* and *KMT2D* seem to both impair the function of highly overlapping GC exit enhancers in B cells. However, it was shown that the simultaneous presence of *CREBBP* and *KMT2D* mutations represent the most highly recurrent, paired gene co-occurrence scenario in B-cell lymphomas[22]. This apparent paradox makes it hard to predict exactly what their combinatorial outcome would be. Formally demonstrating that these mutations cooperate in malignant transformation, and on what basis, is an intriguing question from both the epigenetic and biological standpoints.

Here, we show that combined haploinsufficiency of *Crebbp* and *Kmt2d* in mice cooperatively suppresses super-enhancers driving key B cell-T cell crosstalk related genes, leading to the establishment of an immune-evasive microenvironment manifesting as CD8+ T-cell exhaustion and depletion, and therefore accelerated lymphomagenesis.

## Results

### Combined CREBBP and KMT2D haploinsufficiency accelerates onset of B-cell lymphomas with FL characteristics

The reported co-occurring pattern of *CREBBP* and *KMT2D* mutations across B-cell lymphomas prompted us to further validate those findings in publicly available patient datasets. Examining cohorts of FL and EZB/cluster3 DLBCL (the subset of DLBCL enriched for *CREBBP* and *KMT2D* mutations, $n = 478$ and 319, respectively)[4–6,9,17,22–26], we confirmed significant co-occurrence of these two mutations (p values are 1.68E−6 and 1.77E−18, respectively, Supplementary Fig. 1a). These data led us to explore whether CREBBP and KMT2D loss of function might cooperate to drive malignant transformation of GC B cells. Given that these mutations are generally heterozygous in patients, we focused on generating conditional double heterozygous knockout mice. For this we crossed *Crebbp* and *Kmt2d* floxed conditional knockout mice with the Cγ1-cre strain to induce their heterozygous deletion in GC B cells[27–29]. These animals were further crossed to VavP-BCL2 mice[30] given that *CREBBP* and *KMT2D* mutant FLs and DLBCLs generally harbor *BCL2* translocations[31]. We generated a cohort of mice with the following genotypes: *VavP-BCL2;Cγ1*cre/+*;Crebbp*fl/+*;Kmt2d*fl/+ (**BCL2 + CK**), *VavP-BCL2;Cγ1*cre/+*;Crebbp*fl/+ (**BCL2 + C**), *VavP-BCL2;Cγ1*cre/+*;Kmt2d*fl/+ (**BCL2 + K**), *VavP-BCL2;Cγ1*cre/+ (**BCL2**), and *Cγ1*cre/+ (**WT**) controls (Fig.1a). To expand the number of mice and generate age- and sex-balanced cohorts, bone marrow of donor mice with these engineered alleles was transplanted into lethally irradiated recipients ($n = 35$ per genotype), which were subsequently immunized with the T cell-dependent antigen sheep red blood cells (SRBCs) at several intervals to induce GC formation and lymphomagenesis (Fig. 1a).

A subset of these animals was sacrificed for phenotypic characterization at day 116 post-bone marrow transplant, at which time the first overt illness was observed in mice, or at day 235, when many of the BCL2 + CK mice were reaching their terminal stage ($n = 6-8$ per time point). Cre-mediated heterozygous exon deletion of *Crebbp* and/or *Kmt2d* in the euthanized mice was confirmed by genotyping PCR (Supplementary Fig. 1b, c). At both time points, we observed progressively greater splenomegaly and spleen/body weight ratios in BCL2 + C, BCL2 + K, and BCL2 + CK mice as compared to BCL2 mice, with significant differences observed between BCL2 + CK vs BCL2 animals (Fig. 1b, Supplementary Fig. 1d). Histologic analysis based on H&E and B220 staining of splenic tissue at day 116 showed well-defined although progressively hyperplastic follicular structures containing

small B lymphocytes in BCL2, BCL2 + C and BCL2 + K mice, consistent with low-grade or incipient FL. In contrast, BCL2 + CK mice manifested a mixed picture of aberrant hyperplastic follicles with markedly expanded and distorted lymphoid structures containing intermingled populations of larger B cells and tingible body macrophages, consistent with more aggressive, higher-grade FL (Fig. 1c). There was progressively greater abundance of lymphoma cells infiltrating solid organs in BCL2, BCL2 + C, BCL2 + K, and especially in BCL2 + CK animals (Fig. 1c, Supplementary Fig. 1e). The histologic appearance of BCL2 mice at day 235 showed further enlarged follicles, composed of fairly monotonous small lymphocytes (Supplementary Fig. 1f). Proliferation (Ki67 staining) was relatively modest and was observed in follicular cell clusters (signal in red pulp is non-specific). A similar picture was observed in BCL2 + C animals although with a few larger B cells among malignant follicles. BCL2 + K animals had a small centroblastic or blastoid morphology with greater numbers of proliferating cells. BCL2 + CK manifested highly enlarged and distorted lymphoid structures composed of larger, more proliferative blastoid cells as compared to the other genotypes based on detailed hematopathology reporting (Supplementary Fig. 1f), indicating a histologically more advanced and aggressive FL.

Consistent with the FL phenotype of these tumors, flow cytometry analysis showed progressively increased proportion of GC B cells (CD38−FAS+) among total B cells (B220+) in the four genotypes at day 116, which had plateaued by day 235 (Fig. 1d, Supplementary Fig. 1g–i). Confirming the GC origin of these tumors, there was ample evidence of somatic hypermutation (SHM) at the VDJ VJ558-JH4 locus across all genotypes (Fig. 1e). Of note, the SHM burden was significantly greater in BCL2 + CK lymphoma cells at both time points as compared to all other genotypes (Fig. 1e), suggesting these cells were more exposed to dark zone (DZ)-like proliferative bursting and AICDA activity. Ig isotype composition analysis based on RNA-seq did not reveal bias towards any particular heavy chain selection among all genotypes (Supplementary Fig. 1j). Ig heavy chain VDJ region sequencing indicated that BCL2 + CK lymphomas are more clonal and less diverse based on Simpson clonality or Shannon diversity scores respectively, whereas BCL2 + C and BCL2 + K had an intermediate phenotype as compared to BCL2 lymphomas (Supplementary Fig. 2a, b). No recurrent mutations were identified in whole exome sequencing (WES) analysis of day 235 lymphomas (Supplementary Data 1). Finally, an overall survival analysis showed a similar trend of progressively inferior outcome from BCL2 + C, BCL2 + K, to BCL2 + CK compared to BCL2 (Fig. 1f). Taken together, these data indicate that CREBBP and KMT2D haploinsufficiency cooperates to accelerate development of FLs with less clonal diversity and more aggressive characteristics than either allele alone.

### Combined CREBBP and KMT2D haploinsufficiency induces terminal depletion and exhaustion of CD8+ T cells, preceded by increased TFH/TFR ratios at earlier stages

We next examined whether CREBBP and KMT2D deficiency altered the lymphoma immune microenvironment. CD4+ cell abundance was comparable across all genotypes at both time points (Fig. 1g, Supplementary Fig. 2c). In contrast, there were progressively lower numbers of CD8+ cells that were most statistically significant in the BCL2 + CK setting, especially at the later time point (Fig. 1h, Supplementary Fig. 2c). Further analysis of CD8+ T cell subsets revealed progressive expansion of effector cells (CD8+CD44+CCR7−) with a corresponding reduction of the central memory population (CM, CD8+CD44+CCR7+) at day 116 (Fig. 1i, Supplementary Fig. 2d). Unfortunately, we were unable to accurately separate naive, CM and effector cells at day 235 due to CD62L shedding[32], which was why we switched to CCR7 labeling in day 116 samples. This issue notwithstanding, we noted significant expansion of CD44+ activated CD8+ T cells (including effector and CM populations) in BCL2 + C and BCL2 + CK, whereas the fraction of these

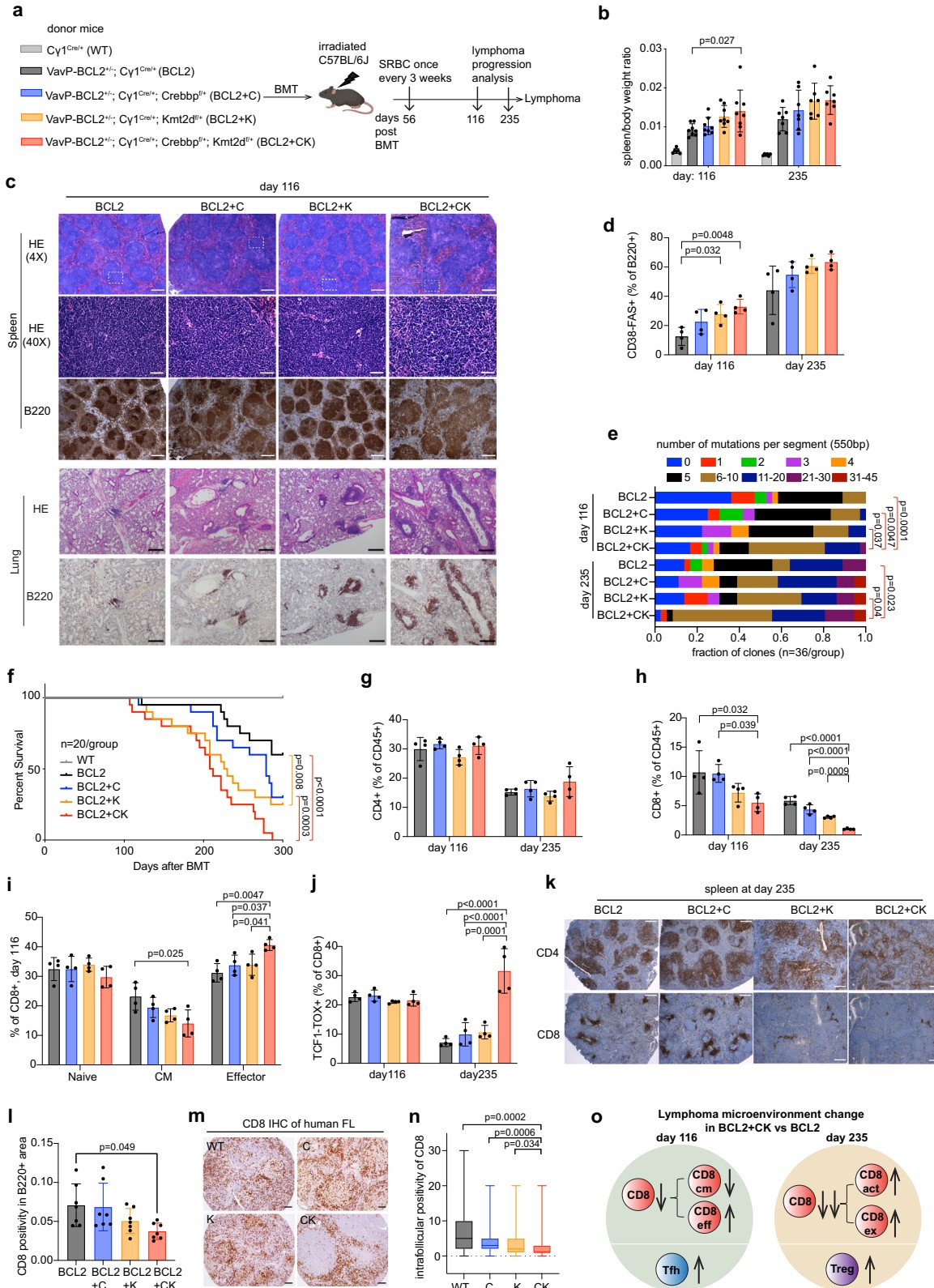

cells in BCL2 + K was reduced (Supplementary Fig. 2e, f). Most remarkably, the proportion of CD8⁺ T cells manifesting the exhausted phenotype (CD8⁺TCF1⁻TOX⁺) was massively increased in BCL2 + CK, but not BCL2 + C and BCL + K, at day 235 (Fig. 1j, Supplementary Fig. 2g), indicating a strong CD8 dysfunction phenotype unique to the BCL2 + CK setting.

Among CD4⁺ cells, T follicular helper (TFH, CD4⁺CXCR5⁺PD1⁺FOXP3⁻) cell frequency was progressively increased across genotypes at both time points, consistent with the pattern of GC B-cell expansion (Supplementary Fig. 2h–j). Intriguingly, TFH expansion occurred out of proportion with GC suppressive T follicular regulatory cells (TFRs, CD4⁺CXCR5⁺PD1⁺FOXP3⁺) at day 116

**Fig. 1 | Combined CREBBP and KMT2D haploinsufficiency accelerates murine lymphomagenesis, featuring a CD8$^+$ T cell-depleted microenvironment.**
**a** Experimental scheme for murine lymphomagenesis. BMT recipients: 8-weeks, female C57BL/6J mice. **b** Mouse spleen/body weight ratio. Mean ± SD, left to right: $n = 6/8/8/8/8$ and $6/7/7/7/7$ mice. **c** Representative H&E and B220 IHC images of mouse spleen and lung sections. Scale bars: 200 pixels in spleen and 380 pixels in lung. **d** FACS analysis showing the frequency of splenic GC B cells (CD38$^-$FAS$^+$). Mean ± SD, $n = 4$ mice per genotype. **e** Mutation burden at IgH-VJ558-JH4 region in mouse lymphoma cells. $n = 36$ clones per genotype. **f** Kaplan−Meier survival curve of the lymphomagenic mice. $n = 20$ mice per group. **g, h** FACS analysis showing the frequency of splenic (**g**) CD4$^+$ and (**h**) CD8$^+$ T cells. Mean ± SD, $n = 4$ mice per genotype. **i, j** FACS analysis showing the frequency of splenic naïve/CM/effector CD8 at day 116 (**i**) or exhausted CD8 (TCF1$^-$TOX$^+$) at day 116 and 235 (**j**). Mean ± SD, $n = 4$ mice per genotype. **k** Representative CD4 and CD8 IHC images of spleen sections. Scale bar = 200 pixels. **l** IHC based quantification of CD8$^+$ cell frequency within B220$^+$ B cell follicles. Mean ± SD, $n = 7$ mice per genotype. **m** Representative CD8 IHC images of human FL tissue microarrays (TMAs)[35]. Scale bar = 100 pixels. **n** CD8$^+$ cell percentage among overall cellularity in human FL TMAs (left to right: $n = 23, 43, 53, 109$). Kruskal−Wallis test followed by Dunn's multiple comparisons test. The box middle line marks the median. The vertical size of box denotes the interquartile range (IQR). The upper and lower hinges correspond to the 25th and 75th percentiles. The upper and lower whiskers extend to the maximum and minimum values that are within 1.5 × IQR from the hinges. **o** A summary of lymphoma microenvironment change in BCL2 + CK vs BCL2. Up and down arrows indicate population increase or decrease respectively in BCL2 + CK vs BCL2. CD8cm/eff/act/ex: central memory/effector/activated/exhausted CD8. P values were determined using ordinary one-way ANOVA followed by Tukey−Kramer's multiple comparisons test (**b, d, g–j, l**), two-tailed Wilcoxon rank sum test (**e**), two-tailed log-rank test (**f**). Source data are provided as a Source Data file.

(Supplementary Fig. 2k, l). Given that TFRs are known to terminate the GC reaction in part by suppressing TFH functions[33,34], we speculate that increased TFH/TFR ratios may favor initial expansion of malignant GC-like structures. Consistently, TFH/GCB ratios were significantly increased in BCL2 + K and BCL2 + CK at day 116 (Supplementary Fig. 2m). Finally, there was an equivalent increase in regulatory T cells (Tregs, CD4$^+$FOXP3$^+$) among mutant genotypes in the end-stage lymphomas (Supplementary Fig. 2n, o), suggesting additional layers of immune dysfunction.

Next, to determine the spatial distribution of T cells, we performed CD4 and CD8 immunohistochemistry on late-stage lymphomas. CD4$^+$ cells exhibited similar abundance, whereas their distribution became progressively more disseminated across the mutant lymphomas (Fig. 1k). In contrast, quantification of CD8$^+$ cell numbers within B cell areas revealed progressive reduction that was most profound in BCL2 + CK mice (Fig. 1k, l). To determine whether a similar phenomenon might occur in humans we examined CD8$^+$ T cell staining patterns using a tissue microarray representing 211 FL patient specimens[35]. These studies showed significant exclusion of CD8$^+$ cells from malignant follicles in CK as compared to C, K or epigenetic WT patients (Fig. 1m, n). Together, these data suggest that CREBBP and KMT2D haploinsufficiency cooperatively remodels the immune microenvironment from a TFH-enriched pro-GC reaction early stage to a CD8-exhausted and depleted later stage to facilitate lymphomagenesis (Fig. 1o).

## CREBBP and KMT2D haploinsufficiency cooperatively induces GC hyperplasia and superior GC B cell fitness
To better understand how loss of CREBBP and KMT2D set the stage for lymphoma formation with the observed phenotypes, we generated mice with the following genotypes: $C\gamma1^{cre/+}$, $C\gamma1^{cre/+};Crebbp^{fl/+}$, $C\gamma1^{cre/+};Kmt2d^{fl/+}$, and $C\gamma1^{cre/+};Crebbp^{fl/+};Kmt2d^{fl/+}$, hereafter referred to as WT, C, K, and CK respectively. Sex and age-matched littermates were immunized with SRBCs to trigger GC formation and were euthanized 10 days later, when GCs reached their peak size (Fig. 2a). Flow cytometry analysis revealed that the abundance of total B cells (B220$^+$) was comparable among all genotypes (Fig. 2b). In contrast, the proportion of GC B cells (CD38$^-$FAS$^+$) was progressively increased, with CK mice manifesting more severe GC hyperplasia compared to either single mutant alone (Fig. 2c, d). To further confirm this phenotype, we performed immunofluorescence (IF) staining of spleen sections using B220-AF488 and Ki67-AF594, which co-label the actively proliferating GC B cells (Fig. 2e). Image quantification showed significant increase of GC size in CK mice, with C and K manifesting intermediate sizes (Fig. 2f). By contrast, there was no change in the abundance of GC structures (Fig. 2g). Among GC B cells, the ratio of centroblasts (CB, CXCR4$^{hi}$CD86$^{lo}$) to centrocytes (CC, CXCR4$^{lo}$CD86$^{hi}$) was significantly skewed towards centroblasts in C and CK (Figs. 2c, h).

We next evaluated whether CK GC B cells exhibit fitness advantage over WT controls when placed within the same microenvironment. For this, chimeric mice carrying equal ratios of WT and CK bone marrow cells were immunized with SRBCs and euthanized 10 days later, 1 h after injection of EdU (Fig. 2i). As expected, WT and CK-derived cells were equally represented in the total B cell population (Fig. 2j-k). By contrast, normalizing the percentage of WT or CK in GC B cells to their respective percentage in total B cells, we observed a significantly increased proportion of CK-derived GC B cells vs WT controls (Fig. 2j, l). This increased competitiveness was not due to changes in proliferation rate as shown by EdU incorporation (Fig. 2j, m). Hence, simultaneous monoallelic deficiency of *Crebbp* and *Kmt2d* confers a fitness advantage to GC B cells without altering cell proliferation rate.

## CREBBP and KMT2D haploinsufficiency induces cooperative disruption of GC transcriptional programming
To better understand these GC and lymphoma phenotypes, we performed RNA-seq profiling of CB and CC sorted from SRBC-immunized mice (Fig. 3a). Both principal component analysis (PCA) and hierarchical clustering analysis revealed a clear segregation of CK CBs and CCs from other genotypes (Fig. 3b, c). We next compared each mutant genotype to WT to define differentially expressed genes among CBs and CCs. This analysis revealed significantly greater perturbation among CK cells, with transcriptional changes skewed towards repression (295 genes up and 1147 genes down in CB, 367 genes up and 1417 genes down in CC, $q < 0.01$ and $|FC| > 1.5$). Many of these genes were also perturbed by C or K, albeit to a generally lower degree (Fig. 3d, e, Supplementary Fig. 3a, b, and Supplementary Data 2, 3). A hypergeometric pathway analysis enriched for gene signatures similarly altered in CK CBs and CCs (Fig. 3f, Supplementary Data 4). In particular, gene sets associated with GC exit signaling, DNA damage repair, and tumor suppressors were expressed at lower levels in CK (Fig. 3f). Conversely, those related to biosynthetic metabolism that enable CCs to return to the dark zone were upregulated in CK (Fig. 3f).

To further study how GC B cell phenotypic transitions are perturbed, we conducted trajectory analysis of the gene expression change from CB to CC for each genotype, and then grouped these differentially expressed genes into eight distinct clusters by Fuzzy c-means clustering[36] (Supplementary Fig. 3c, Supplementary Data 2, 3). Trajectories 3 (Traj_3) and 4 (Traj_4) were most informative. Traj_3 genes (e.g., *Cd86* and *H2-Oa*) are normally upregulated when CBs transition to CCs, and displayed progressively lower basal expression in CBs and impaired upregulation in CCs across C, K and CK (Fig. 3g). Reciprocally, Traj_4 genes (e.g., *Cxcr4* and *Ccnb1*) that are repressed in WT CCs vs CB, were expressed at higher baseline levels and were progressively less repressed across C, K and CK genotypes (Fig. 3g).

In agreement, pathway analysis indicated that Traj_3 was significantly enriched with CC-signature immune synapse genes linked to B/T crosstalk such as NFkB, BCR, IL10, IL4 and STAT3 signaling (Fig. 3h, Supplementary Data 4). On the contrary, Traj_4 was enriched with

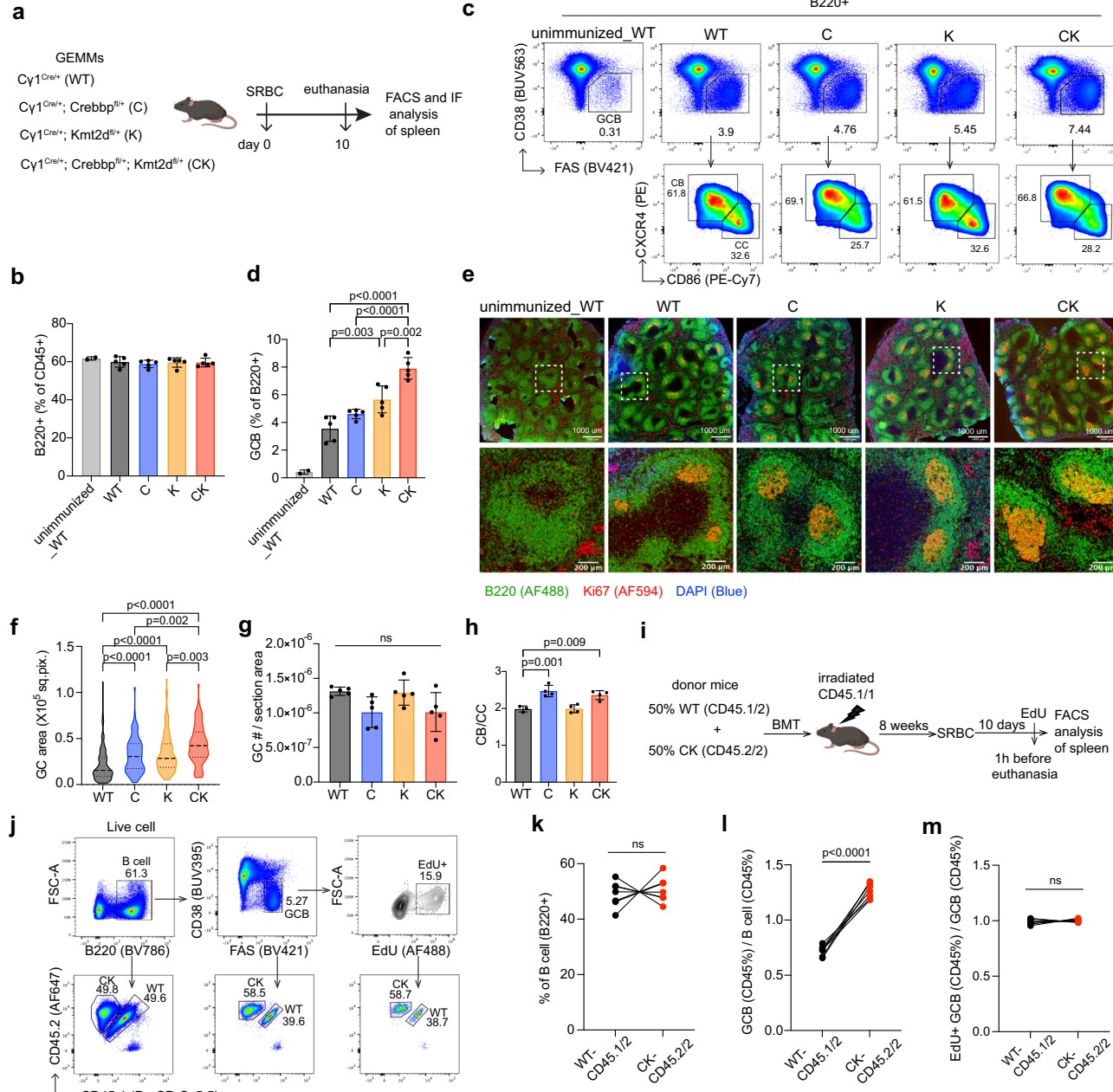

**Fig. 2 | CREBBP and KMT2D haploinsufficiency cooperatively induces hyperplastic GCs with superior fitness. a** Experimental scheme for GC characterization (results shown in **b**–**h**). 8-weeks, female C57BL/6J background mice were used. **b**, **d**, **h** FACS analysis showing the frequency of splenic total B cells (**b**, B220⁺), GC B cells (**d**, CD38⁻FAS⁺), and (**h**) CB (CXCR4ʰⁱCD86ˡᵒ) / CC (CXCR4ˡᵒCD86ʰⁱ) ratios. Mean ± SD, left to right: *n* = 2/5/5/5/5 (**b**, **d**) or 3/4/4/4 (**h**) mice. **c** Representative FACS plots showing the gating strategy and relative frequency of splenic GC B cells, CB and CC. **e** B220 (green), Ki67 (red), and DAPI (blue) IF images of spleen sections. Bottom images show the zoom-in of outlined areas in top images. Scale bars: 1000 um (top), 200 um (bottom). **f**–**g** Quantification of (**f**) GC area (left to right: *n* = 119, 91, 122, 95 GCs) and (**g**) relative GC number normalized to spleen section area (mean ± SD, *n* = 5 mice per genotype). **i** Experimental design for fitness study (results shown in **j**–**m**). BMT recipients: 8-weeks, female B6.SJL mice.

**j** Representative FACS plots showing the gating strategy and relative frequency of indicated splenic cell types (left to right: total B, GC B, EdU+ GC B cells). WT and CK-derived cells were separated as CD45.1/2 and CD45.2/2, respectively. **k** FACS data showing the proportion of WT and CK-derived splenic total B cells. Each pair of connected dots represents a mouse (*n* = 7 mice). **l**, **m** FACS data showing the ratio of WT or CK-derived GC B cell percentage to their respective parental total B cell percentage (**l**) or EdU+ GC B cell percentage to their respective parental total GC B cell percentage (**m**). Each pair of connected dots represents a mouse (*n* = 7 mice). P values were determined using ordinary one-way ANOVA followed by Tukey-Kramer's post-test (**d**, **g**, **h**), Kruskal–Wallis test followed by Dunn's multiple comparisons test (**f**), two-tailed paired Student's *t* test (**k**–**m**). Source data are provided as a Source Data file.

genes involved in CBs and their classical G2/M cell cycle progression phenotype. We validated the reduction of these CC signature genes in independent experiments using RT-qPCR (Fig. 3i). Taken together, these results demonstrate that CREBBP and KMT2D haploinsufficiency cooperatively attenuates induction of genes essential for T-cell directed GC exit signaling.

## CREBBP and KMT2D haploinsufficiency skews GC B cell fate decisions towards DZ vs GC exit

To evaluate how intra-GC cell state transitions are perturbed, we performed single-cell RNA-seq profiling of sorted splenic B220⁺IgD⁻ B cells from each genotype at day 10 post-SRBC immunization. Uniform manifold approximation and projection (UMAP) was

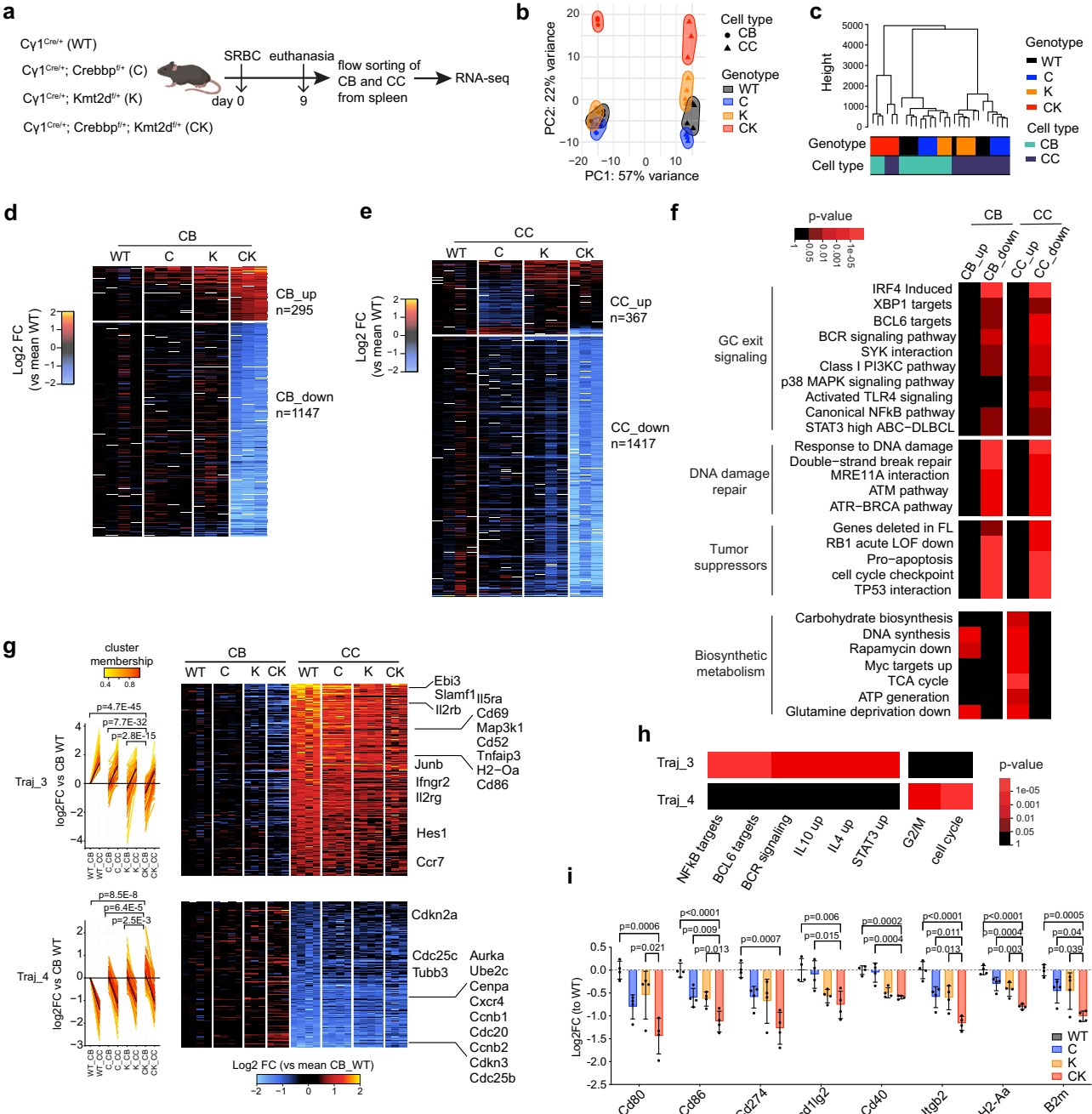

**Fig. 3 | RNA-seq reveals cooperative perturbation of intra-GC transcriptional transitions by CREBBP and KMT2D haploinsufficiency. a** Experimental scheme for RNA-seq profiling of CB and CC (results shown in **b**–**h**). 8-weeks, female C57BL/6J background mice were used. **b** PCA of RNA-seq datasets. WT/C/K/CK: $n = 4/4/3/3$ mice (CB), $n = 4/4/4/3$ mice (CC). **c** Dendrogram showing the hierarchical clustering result of RNA-seq datasets using the top 10% most variable genes, the Manhattan distance and ward.D2 linkage. **d**, **e** Heatmap showing the relative expression levels of the union differentially expressed genes (DEGs) as log2FC (vs mean WT expression) for each genotype in (**d**) CB and (**e**) CC. Union DEGs include DEGs defined in at least one pair-wise comparison using WT as control with a significance cut-off of padj < 0.01, |log2FC| > 0.58. Scale factors, based on single-cell RNA-seq UMI counts, were applied to account for total mRNA difference. Each column represents one mouse dataset. **f** Pathway enrichment analysis for CK vs WT DEGs using Parametric Analysis of Gene Set Enrichment (PAGE). **g** Fuzzy c-means clustering of RNA-seq datasets identified 8 clusters (named as Traj_1 to Traj_8) with distinct trajectory patterns, Traj_3 and Traj_4 are shown: line plot (left) and heatmap (right) of log2FC expression (vs mean of WT CB). Black lines in the line plot are cluster centroid; genes are colored by the degree of cluster membership. A linear regression model was fit for log2FC expression compared to WT as a function of cell type (CB or CC), genotype (C, K, or CK), and interaction of cell type and genotype. The corresponding *p* values for the coefficients are shown. **h** Pathway enrichment analysis using PAGE for Traj_3 and Traj_4 genes. **i**, RT-qPCR of indicated genes in sorted GC B cells for each genotype ($n = 4$ mice per genotype). qPCR signal was normalized as log2FC (vs mean WT) and presented as mean ± SEM. *P* values were calculated by one-tailed hypergeometric test (**f**, **h**) and ordinary one-way ANOVA followed by Tukey–Kramer's multiple comparisons test (**i**). Source data are provided as a Source Data file.

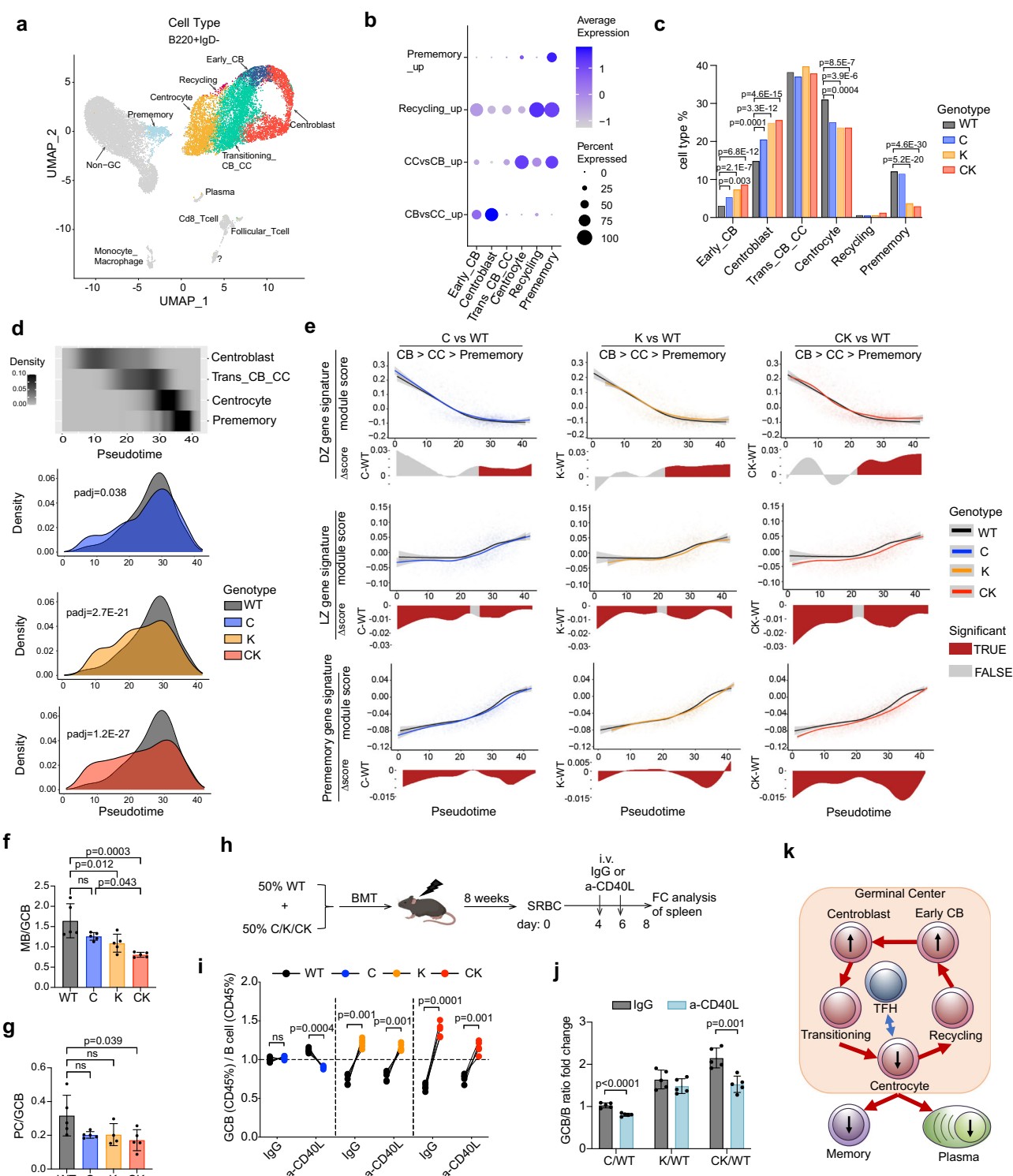

applied for dimensionality reduction, then cell subpopulations were identified using graph-based clustering and K nearest neighbor (Fig. 4a). No sample specific batch effects were observed (Supplementary Fig. 4a). GC B cell clusters were subsequently annotated as early_centroblast (early_CB), centroblast, transitioning_CB_CC (trans_CB_CC), centrocyte, recycling, and prememory by label transfer from previously published data sets[37]. The accuracy of these assignments was further validated using cell type-specific gene signatures and marker genes (Fig. 4a-b, Supplementary Fig. 4b). Differential cell abundance analysis revealed that the proportion of early_CBs and CBs were progressively increased,

whereas that of centrocytes and prememory B cells were progressively decreased across the genotypes (Fig. 4c).

To further assess these changes, we used slingshot[38] to generate a pseudotime trajectory starting in CB, passing through the trans_CB_CC and CC, and ending at prememory cells (Fig. 4d). Plotting cell density for each genotype along this pseudotime axis showed progressively more significant shifts in distribution towards the CB side in C, K and CK (Fig. 4d). We next projected DZ, light zone (LZ), and prememory gene signature scores (module scores) along the pseudotime axis and assessed whether their pseudo-temporal expression patterns are altered. Notably, DZ signature genes were aberrantly maintained at

**Fig. 4 | CREBBP and KMT2D haploinsufficiency cooperatively skews the GC B cell fate toward CB over exit differentiation into MB and PC. a**–**e** Single-cell RNA-seq profiling of splenic B220$^+$IgD$^-$ cells sorted from SRBC-immunized mice (8-weeks, female, C57BL/6J background) at day 10. **a** UMAP plot showing identified cell types, with GC B cell subtypes color highlighted. **b** Seurat dot plot showing the average expression and percent of cells expressing the indicated gene signatures in different GC B subtypes. **c** Bar plot depicting the proportion of different GC B subtypes in each genotype. **d** Cell density plot showing the distribution of different GC B subtypes along a Slingshot pseudotime axis anchored at CB. **e** Top, gene signature module scores were plotted for each cell along the pseudotime axis, with a best fit spline curve representing the average score. Gray bands represent 95% confidence intervals. Bottom, differential expression was shown as a delta spline plot across pseudotime and colored based on statistical significance for each bin. **f**, **g** FACS analysis showing the relative abundance of MB (**f**, *n* = 5 mice per genotype)

or PC (**g**, left to right: n = 5/5/4/5 mice) vs GC B cells at day 10 post-SRBC (mean ± SD). **h** Experimental scheme for CD40L blocking assay (results shown in **i**, **j** *n* = 5 mice per group). **i** FACS data showing the ratio of WT, C, K or CK-derived GC B cell percentage to their respective parental total B cell percentage in either control IgG or CD40L blocking antibody treated mice. Each pair of connected dots represents a mouse, two-tailed paired Student's *t* test. **j** GCB (CD45%) / B cell (CD45%) ratio fold change (i.e., C/WT, K/WT, and CK/WT) in either control IgG or CD40L blocking antibody treated mice, mean ± SD, two-tailed unpaired Student's *t* test. **k** Graphical representation depicting the cell state transitions within GC. Upward and down-ward black arrows indicate cell abundance increase and decrease respectively in CK vs WT. *P* values were determined using two-tailed Fisher's exact test (**c**), two-tailed Wilcoxon rank sum test (**d**, **e**), ordinary one-way ANOVA followed by Tukey–Kramer's post-test (**f**, **g**). Source data are provided as a Source Data file.

higher levels in CC and prememory cells from C, K and to a greater degree CK (Fig. 4e). Conversely, LZ and prememory signature genes were expressed at lower levels in C, K and to a greater degree CK, starting at CB and persisting throughout the entire differentiation trajectory, implying that C, K, and especially CK-deficient GC B cells fail to properly engage GC exit genes and thus manifest impaired GC exit capabilities (Fig. 4e). Indeed, measuring abundance of memory and plasma cells relative to GC B cells by flow cytometry revealed significant reduction in these populations that was most severe in CK-deficient mice (Fig. 4f, g, Supplementary Fig. 4c, d).

We next explored whether CK deficiency affected antibody affinity maturation and long-lived plasma cells (LLPCs) abundance by analyzing NP-OVA immunized mice (Supplementary Fig. 4e). ELISA assays showed significant reduction of both low (NP28) and high (NP8) affinity NP specific IgG1 at 70 days in CK mice, although the ratios of high to low affinity IgG1 was not changed (Supplementary Fig. 4f–h). Accordingly, ELISPOT showed a trend towards reduced number of NP8 reacting IgG1 secreting LLPCs in CK bone marrow (Supplementary Fig. 4i, j). This was not due to defective class-switching, since on the contrary IgG1-switched GC B cells were more abundant in CK (Supplementary Fig. 4k–m). Given that class-switching and affinity maturation were unperturbed, we reasoned that reduction in high-affinity IgG1 secreting cells was caused by GC exit impairment.

The GC B cell fate perturbations led us to examine whether CK GC B cells were still dependent on TFH help. We therefore generated 3 groups of bone marrow chimeric mice (WT + C, WT + K, WT + CK), immunized with SRBC, and administered CD40L-blocking antibodies or control IgG on days 4 and 6 (after GCs had formed) (Fig. 4h). In IgG control mice, C exhibited similar fitness as WT, whereas K and to a greater extent CK conferred a significant fitness advantage (Fig. 4i, j). Notably, upon CD40L blockade, the competitive advantage of CK GC B cells was significantly reduced. In contrast, fitness advantage of K GC B cells was only mildly reduced, whereas C GC B cells displayed a small but significant fitness disadvantage (Fig. 4i, j, Supplementary Fig. 4n). Therefore, the CK phenotype manifests a combination of distinct immune synapse related perturbations such as fitness advantage, which is primarily K related, and TFH dependency, which is primarily C related.

Collectively, these data suggest that CREBBP and KMT2D haploinsufficiency cooperatively reprograms T-cell dependent GC B cell fate decisions, altering their phenotypic transitions to favor the DZ mutagenic state, while impairing commitment to memory or plasma cell fates (Fig. 4k). This effect aligns with the heavier somatic hyper-mutation burden observed in CK lymphomas vs controls (Fig. 1e).

## CREBBP and KMT2D haploinsufficiency cooperatively blocks dynamic activation of enhancers required for intra-GC cell state transition

To explain CK haploinsufficiency-induced gene expression perturbation, we performed ATAC-seq profiling of GC B cells purified from SRBC-immunized mice. Both PCA and unsupervised hierarchical

clustering revealed clear segregation of the four genotypes (Fig. 5a, b). In these analyses C was clearly different from WT, whereas K and CK were highly distinct to WT and C, but more related to each other. We next identified differentially accessible peaks in each mutant genotype relative to WT. K-means clustering grouped these peaks into three distinct clusters: K_CK_Loss (*n* = 218) manifested similar reduction in K and CK vs WT and C, while CK_Gain (*n* = 1153) and CK_Loss (*n* = 828) showed progressive gain or loss of signal respectively from C, K to CK relative to WT (Fig. 5c, d, Supplementary Data 5). Genomic feature annotation revealed that CK_Gain and K_CK_Loss peaks were significantly enriched at promoters and enhancers respectively, whereas CK_Loss peaks were enriched at both enhancers and super-enhancers (Fig. 5e). Accordingly, binning peaks based on their accessibility fold change showed progressively greater representation of enhancers at sites with greater loss of ATAC-seq reads (Fig. 5f).

We next performed integrated RNA-seq and ATAC-seq analysis using GSEA, which indicated that CK_Loss target genes (*n* = 896, annotated by closest TSS using GREAT) were expressed at significantly lower levels in CK CBs and CCs (Fig. 5g, Supplementary Fig. 5a). Conversely, topologically associating domains (TADs)[39] containing CK-repressed genes were enriched with closing ATAC-seq peaks (Fig. 5h, i), consistent with the general co-localization of enhancers with their target genes within the same TADs. Examining transcription factor (TF) motifs underlying these closing peaks yielded enrichment for immune synapse responsive TFs such as SPIB, SPI1 and STAT3[40,41] (Fig. 5j). Expression levels of these TFs were uniform across the genotypes (Supplementary Fig. 5b), ruling out the possibility that their expression changes caused the enrichment.

Finally, to gain more insight into the relationship of CK-regulated enhancers with CC transcriptional programs, we performed t-SNE dimensionality reduction and K-means clustering analysis of union peaks, generated by taking the consensus peaks from our ATAC-seq datasets, published mouse GC B cell H3K27ac[39], H3K4me1, H3K4me3 Mint-ChIP and CB/CC ATAC-seq datasets. These union peaks segregated into eight distinct clusters (Fig. 5k). Notably, cluster 1 peaks gained accessibility in CCs vs CBs, and became progressively less accessible from C, K to CK relative to WT (Fig. 5k, l, Fig. 5m, ATAC column). The respective genes (based on GREAT) were likewise upregulated in CCs vs CBs and progressively repressed in C, K and CK (Fig. 5m, RNA column). In normal GC B cells these sites were generally marked by H3K27ac, H3K4me1 and H3K4me3, and located quite distal from TSS, suggesting that they correspond largely to enhancers (Fig. 5m, distance to TSS column). In agreement, TADs containing Traj_3 genes showing upregulation in CCs were enriched with closing ATAC-seq peaks in CK (Fig. 5n). Overall, CREBBP and KMT2D haploinsufficiency cooperatively and selectively restricts dynamic activation of enhancers governing CB-to-CC cell state transition. Such selectivity may be conferred by specific sensitivity of certain GC cell fate-determining TFs to the dosage reduction of CREBBP and KMT2D.

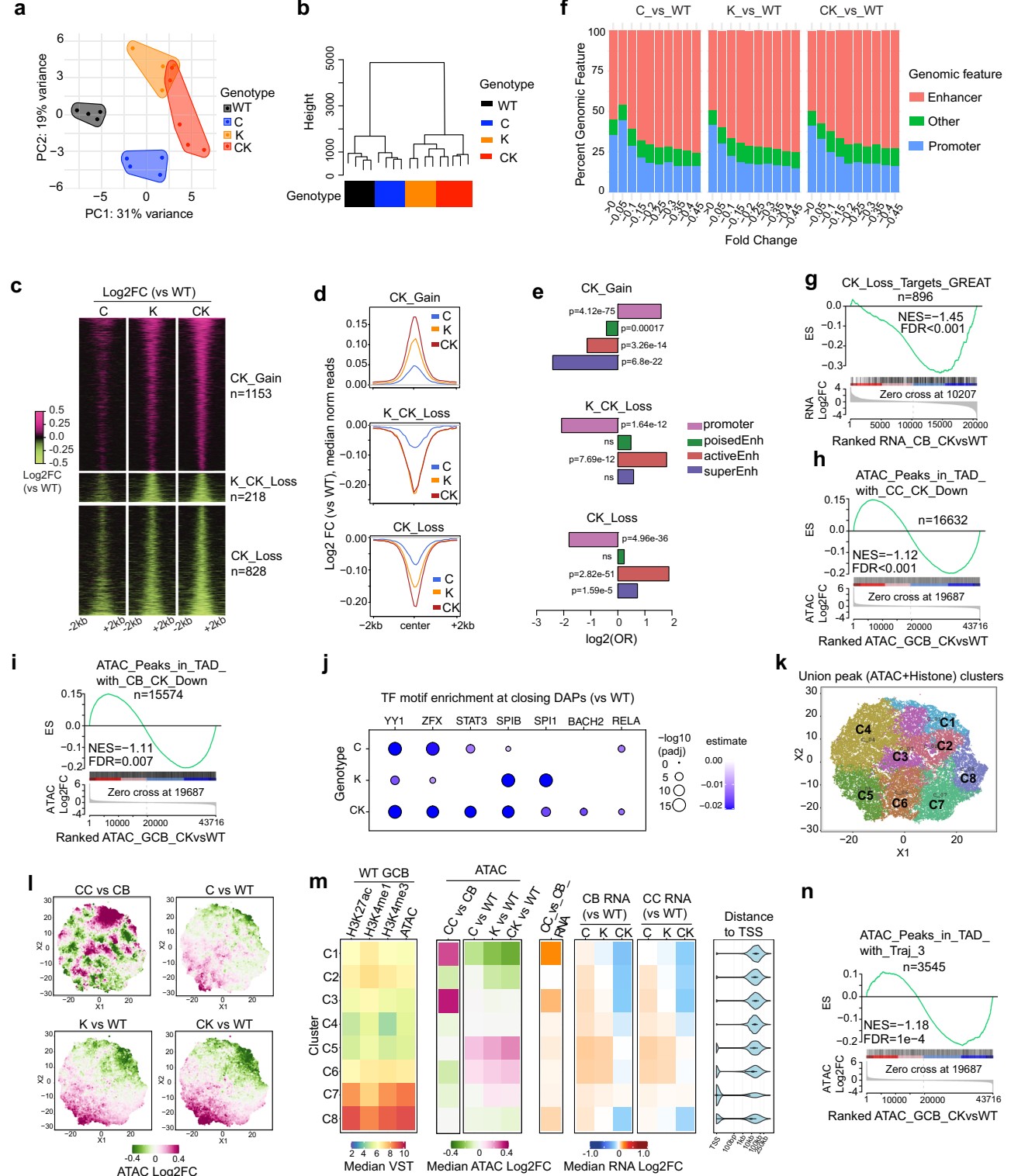

## CREBBP and KMT2D form a complex, with interdependent chromatin modifying functions at immune synapse gene enhancers

To further explore the nature of CREBBP and KMT2D cooperation in human lymphoma cells, we generated several isogenic clones of the GCB-DLBCL cell line OCI-Ly7 as follows: *CREBBP*[R1446C] (C, a HAT inactivating mutation often found in human lymphomas), *CREBBP*-KO (CKO), *KMT2D*-KO (K), *CREBBP*[R1446C] + *KMT2D*-KO (CK), as well as CRISPR non-edited control clones (WT) (Supplementary Fig. 6a–e). RNA-seq of these isogenic OCI-Ly7 lines revealed that CK perturbed

transcriptome to a greater extent (PC1) and in a different manner (PC2) compared to C or K alone (Fig. 6a). Genes downregulated ($n = 1672$) or upregulated ($n = 1282$) in CK cells were also expressed at lower or higher levels respectively in CK-mutant primary human GCB-DLBCLs (vs epigenetic WT lacking C, K or EZH2 mutations) (Fig. 6b, Supplementary Fig. 6f). Reciprocally, the same was observed when performing GSEA on the set of genes either down or up regulated in CK-mutant primary human GCB-DLBCL vs the ranked isogenic cell gene list (Supplementary Fig. 6g, h). Hence our isogenic cells reflected the perturbations observed in primary DLBCL cases.

**Fig. 5 | CREBBP and KMT2D haploinsufficiency cooperatively disrupts dynamic enhancer accessibility remodeling required for intra-GC cell state transitions.** **a, b** PCA plot (**a**) and Dendrogram showing the hierarchical clustering (**b**) of mouse GC B cell ATAC-seq datasets. WT/C/K/CK: *n* = 4/4/4/5 mice (8 weeks, female, C57BL/6J background). **c** K-means clustered heatmap showing the relative ATAC-seq read density of the union differentially accessible peaks (DAPs). Union DAPs include DAPs (padj < 0.01) defined in at least one pair-wise comparison using WT as control. **d** Aggregate (median) ATAC-seq read intensity plot around peak center (±2kb) for each cluster. **e** Odds ratio (OR)-based association analysis between each DAP cluster and indicated genomic features. **f** Genomic feature distribution of indicated DAP bins. **g** GSEA using CK_Loss target genes (GREAT) as gene set against a ranked CB RNA-seq gene list (CK vs WT). NES, normalized enrichment score. **h, i** GSEA using ATAC-seq peaks in TADs containing CK downregulated genes in (**h**) CC or (**i**) CB as peak set against ranked ATAC-seq peak list (CK vs WT). **j** Dot plot showing TF motifs enriched at closing ATAC-seq peaks in C, K or CK relative to WT (*n* = 1125,

4393, 10777 peaks respectively). A multivariate TF regulatory potential model was employed to identify the enriched TFs. Dots are colored by accessibility remodeling score and sized by −log10(padj). **k** t-SNE plot showing eight distinct clusters (C1–C8) among union histone mark and ATAC-seq peaks. **l** t-SNE plot of indicated relative (log2FC) ATAC-seq read density. **m** Heatmaps showing median read density of the indicated histone marks or ATAC-seq (left), or relative (median log2FC) ATAC-seq (middle) and RNA-seq (right) signal for each cluster. Distance to TSS plot shows the distance of union peaks to their closest TSSs. **n** GSEA using ATAC-seq peaks in TADs containing Trajectory_3 genes (Fig. 3g) as peak set against ranked ATAC-seq peak list (CK vs WT). *P* values were calculated by two-tailed Fisher's exact test (**e**), an empirical phenotype-based permutation test and adjusted for gene set size and multiple hypotheses testing (**g–i, n**), two-tailed Student's *t* test and BH-adjusted for multiple comparisons (**j**). Source data are provided as a Source Data file.

To examine the impact of these mutant genotypes on chromatin, we first conducted western blots for H3K27ac and H3K4me1 in our isogenic cells and observed no significant difference (Supplementary Fig. 6i–k). This did not preclude perturbation of the distribution of these marks throughout the genome, and therefore we also performed CUT&RUN (Cleavage Under Targets & Release Using Nuclease) profiling of H3K27ac, H3K4me1 and H3K4me3. For the unbiased analysis of these data, we performed t-SNE dimensionality reduction and K-means clustering of union consensus peaks from all histone marks, which defined eight distinct clusters (Fig. 6c, d). Cluster 1 loci manifested progressive reduction of H3K27ac and H3K4me1 but not H3K4me3 in C, K and CK, their corresponding target genes were progressively repressed, and these loci were also distal to TSSs, collectively suggesting that these regions corresponded mostly to gene enhancers (Fig. 6e, f). Adjacent clusters 2 and 3 appeared as less affected versions of cluster 1, whereas clusters 4−6 represented gene promoters. Clusters 7−8 were linked to genes strongly upregulated in all mutant genotypes, of unclear significance from the chromatin perspective. Cluster 1 loci/genes thus appear to be most directly dependent on both CREBBP and KMT2D.

Unexpectedly, plotting read density of cluster 1 peaks showed that single loss of either CREBBP or KMT2D resulted in reduction of both H3K27ac and H3K4me1 (Fig. 6g, h, Supplementary Data 6). To better understand this phenomenon, we analyzed ChIP-seq for CREBBP and KMT2D in OCI-Ly7 cells[18,42], which showed that 55% of KMT2D peaks overlapped with CREBBP, whereas 38% of CREBBP peaks overlapped with KMT2D (Fig. 6i). Regardless of overlap, KMT2D and CREBBP peaks mapped mainly to intergenic, distal and intronic elements, consistent with their predominant enhancer-related function (Fig. 6j). This interdependency led us to explore whether these two proteins could form a complex. We therefore performed endogenous protein co-immunoprecipitations in two independent GCB-DLBCL cell lines (OCI-Ly7 and SUDHL4), and observed reciprocal enrichment for each protein with their respective antibodies but not with control antibody (Fig. 6k, Supplementary Fig. 6l), with a higher fraction of KMT2D associated with CREBBP than vice versa, consistent with ChIP-seq analysis (Fig. 6i). We reasoned that such interaction might contribute to their stable chromatin association. Indeed, ChIP-qPCR assays revealed that loss of CREBBP and KMT2D impaired each other's stable association to CK target enhancers (Fig. 6l, m).

Next, to better understand the transcriptional programs and biological functions linked to cluster 1 (i.e., the most CK dependent genes), we performed Enrichr and ToppGene analyses[43,44]. These analyses showed significant enrichment for TFs (e.g., SPI1, RELA, RELB, STAT3) and pathways (e.g., BCR, TLR, IFNγ, CTLA4 blockade) implicated in CC immune synapse signaling (padj<0.05, Fig. 6n, Supplementary Data 3). These genes were further associated with defective immune reaction phenotypes such as abnormal CD8 and CD4 responses. We further validated that CK loss induced more severe

repression than either loss alone of key B-cell/T-cell immune synapse target genes in independent experiments at both the mRNA (RT-qPCR) and protein (FACS) levels (Supplementary Fig. 6m–o).

Finally, this strong downregulation of key B-cell/T-cell crosstalk-related genes prompted us to check whether CK deficiency impaired lymphoma cell-mediated CD8 activation. For this, we pulsed isogenic OCI-Ly7 (WT or CK) with either vehicle (DMSO) or HLA class I peptide (CEF), then irradiated and co-cultured them with HLA-A matched human CD8+ cells in vitro (Fig. 6o). FACS analysis showed that CK cells were defective in activating CD8+ T cell expansion and differentiation in both DMSO and CEF-treated settings, as evidenced by the increase of naïve (hCD45RA+hCD62L+hCD95−) and CM (hCD45RA−hCD62L+), and concordant decrease of EM (effector memory, hCD45RA−hCD62L−) or effector (hCD45RA+hCD62L−) CD8+ cells upon CK stimulation vs WT (Fig. 6p, Supplementary Fig. 6p-s). Along these lines, restimulation of CK-exposed CD8+ cells yielded reduced induction of IFNγ and TNFα (Fig. 6q, Supplementary Fig. 6t–v), two signature effector cytokines of cytotoxic CD8+ cells involved in tumor clearance[45].

In summary, these results indicate that CREBBP and KMT2D co-occupy inducible enhancers of B-cell/T-cell immune synapse/GC exit cell fate determination pathways, where they enhance each other's stable chromatin loading and hence histone modification activity, with their simultaneous deficiency eliciting a more profound perturbation of these enhancers than either single deficiency.

## Super-enhancers are particularly susceptible to CREBBP and KMT2D deficiency-induced perturbation

Cell fate determination is proposed to depend on super-enhancers to induce expression of phenotype-driving genes[46]. Given that CK dependent enhancers were enriched for such genes and pathways, and were located quite distally to TSSs, we wondered whether their co-depletion might disproportionately affect super-enhancers. Comparing differential accessibility at enhancers and super-enhancers based on our murine GC B cell ATAC-seq data revealed that super-enhancers were clearly more severely closed, especially in CK setting (see 95% confidence intervals, Fig. 7a). We then summed constituent accessibility peaks for each super-enhancer and ranked them based on their log2 fold change vs WT (Supplementary Fig. 7a−c), which showed a progressively elevated fraction of closing super-enhancers from C, K to CK. Importantly, target genes of these closing super-enhancers included many B-cell/T-cell immune synapse response genes, as exemplified by *Cd86*, *Il9r* and *Zbtb20* (Fig. 7b), which were most strongly downregulated in CK-deficient CB and CC (Supplementary Fig. 7d, e).

Likewise, H3K27ac reduction is more severe at the constituent peaks of super-enhancers than regular enhancers among mutant OCI-Ly7 cells, with the reduction at CK super-enhancers being most dramatic (Fig. 7c). We next focused on regulatory elements with ChIP-seq confirmed binding of both CREBBP and KMT2D, and examined their

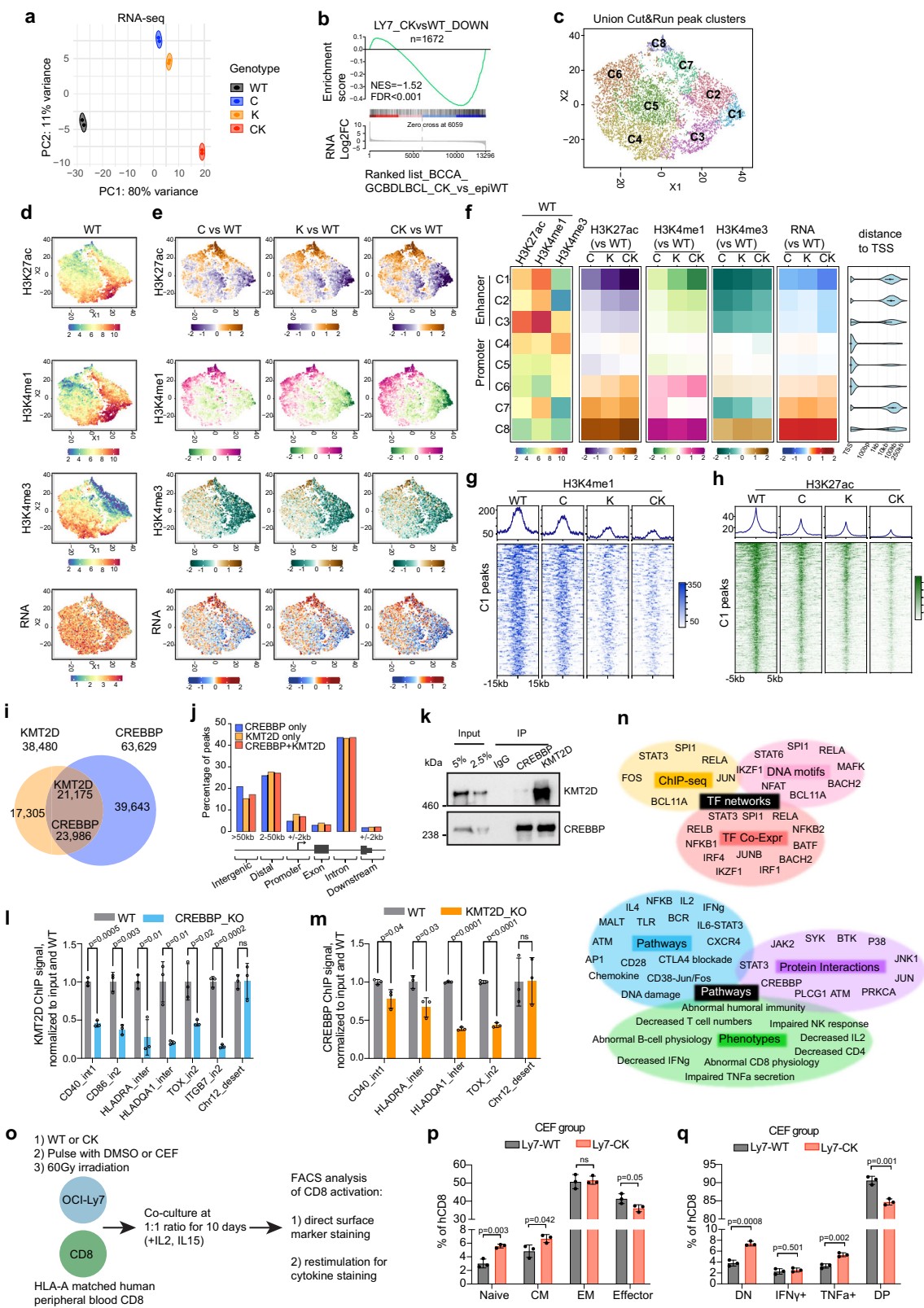

target gene expression. Notably, target genes of CK-bound super-enhancers were more repressed than those of CK-bound regular enhancers and promoters upon loss of C, K and to a substantially greater extent CK (Fig. 7d). Next, we summed constituent H3K27ac peaks for each super-enhancer, and ranked them based on log2 fold change vs WT, and linked them to their corresponding genes within the same human GC TADs[47] (Fig. 7e). GSEA analysis showed that target

genes of super-enhancers bearing reduced H3K27ac tend to be downregulated especially in the CK setting (Fig. 7e).

To further confirm that super-enhancers of interest physically interact with their respective genes, we examined their connectivity based on available H3K27ac HiChIP performed in OCI-Ly7 cells[48]. Plotting the impact of C, K or CK deficiency on super-enhancers that connect with important immune synapse responsive genes such as

**Fig. 6 | CREBBP and KMT2D form a complex, whereby they reciprocally regulate each other's chromatin binding and histone modifying activity. a** PCA plot of isogenic OCI-Ly7 RNA-seq datasets (*n* = 2 per genotype). **b** GSEA using CK-downregulated genes in OCI-Ly7 as gene set against ranked BCCA cohort GCB-DLBCL patients RNA-seq gene list based on CK vs epigenetic WT (epiWT, no mutations in *CREBBP*, *KMT2D*, and *EZH2*). The *p* value was calculated by an empirical phenotype-based permutation test. The FDR is adjusted for gene set size and multiple hypotheses testing. **c** t-SNE plot showing eight distinct clusters (C1-C8) among union CUT&RUN peaks. **d** t-SNE plots showing VST normalized counts of indicated histone marks or RNA in WT OCI-Ly7 cells. **e** t-SNE plots showing read density changes (log2FC, vs WT) for indicated histone marks and RNA. **f** Heatmaps showing median read density (left) or read density changes (median log2FC) of indicated histone marks (middle) or RNA (right) for each cluster. Distance to TSS plot shows the distance of union peaks to their closest TSSs. **g, h** Average signal profiles (top) and heatmaps (bottom) displaying (**g**) H3K4me1 and (**h**) H3K27ac signals at C1 peak regions. **i** Venn diagram displaying overlap between CREBBP and KMT2D ChIP-seq peaks in OCI-Ly7. **j** Genomic feature annotation of CREBBP and KMT2D ChIP-seq peaks. **k** Co-IP showing interaction between endogenous CREBBP and KMT2D in OCI-Ly7. **l, m** ChIP-qPCR of (**l**) KMT2D and (**m**) CREBBP at indicated gene loci in isogenic OCI-Ly7 cells. ChIP signals were normalized to input and then to WT and presented as mean ± SD. **n** Functional annotation of C1 genes (*n* = 401) by Enrichr and Toppgene[43,44]. **o** Experimental design for OCI-Ly7 and human CD8 in vitro co-culture assay. CEF is a pool of HLA class I-restricted virus peptides. **p, q** FACS analysis showing the frequency of different CD8 subtypes (**p**) or cytokine-producing CD8 cells (**q**, DP: IFNγ⁺TNFa⁺, DN: IFNγ⁻TNFa⁻) in CEF-treated co-cultures. Mean ± SD, *n* = 3 wells per co-culture. *P* values were calculated by two-tailed unpaired Student's *t* test and BH-adjusted for multiple comparisons (**l, m, p, q**). Source data are provided as a Source Data file.

*CD86* and *IL21R* further allowed direct visualization of the progressive loss of H3K27ac, H3K4me1 on relevant super-enhancers and the corresponding suppression of gene expression (Fig. 7f). These data suggest that the cooperative effect of CREBBP and KMT2D deficiency on B cell phenotype and lymphomagenesis may be due to a more critical biochemical need for CK complex formation at distal immune synapse responsive super-enhancers.

## CREBBP and KMT2D deficiency suppresses a core immune signature in GC B cells that is retained and shared between murine and human lymphomas

We reasoned that gene expression signatures induced by CK deficiency in GC stage and persisting upon malignant transformation would indicate these effects most likely provide a selective advantage to lymphoma cells. For this, we performed RNA-seq profiling of sorted B220⁺ murine lymphoma cells at day 235, and then identified the set of genes repressed in BCL2 + CK relative to BCL2. These genes were applied to a GSVA-based estimation of their relative expression across an RNA-seq dataset derived from human FL patients (*n* = 16,10,8,15 for epigenetic WT, C, K, and CK, respectively)[17]. This analysis indicated that the CK repressed gene signature expression was progressively decreased from C, K, to CK compared to patients without C, K or *EZH2* mutations (defined as epigenetic WT), with the decrease being significant only for the CK cases (Fig. 8a). We then performed a comparative cross-species analysis of CK repressed signatures in murine CBs, CCs and lymphoma cells, as well as human DLBCL cell lines and FL patients, which identified a list of CK target genes that exhibit consistent downregulation compared to WT in at least two different datasets (Fig. 8b, Supplementary Data 3). Once again functional annotation of this CK_consistent_down gene set recapitulated our findings in isogenic OCI-Ly7 cells, with disruption of numerous critical B-cell/T-cell crosstalk-related TFs, genes and pathways (Fig. 8c). Together, these data point to sustained suppression of B-cell/T-cell immune synapse genes as contributing to transformation, immune evasion from CD8⁺ T cells and maintenance of CK mutant lymphomas.

## Discussion

Herein, we explored the puzzling co-occurrence of somatic mutations in the two enhancer activating chromatin modifier proteins CREBBP and KMT2D in B-cell lymphomas. We confirmed that these mutations are genetically highly co-occurrent in additional FL and GCB-DLBCL cohorts. We found that conditional heterozygous knockout of these chromatin modifiers do indeed cooperate to yield a more aggressive lymphoma phenotype, suggesting that this combination is especially beneficial in providing a selective advantage to pre-malignant B cells. We did not model double homozygous loss of function, since this scenario rarely if ever occurs in humans, and we speculate they may confer an unfavorable functional state to GC B cells. It remains unknown if there is a specific order that these mutations need to occur in the pre-lymphoma state, since presumed clonal precursor B cells have been identified with both of these genetic lesions.

It is well-established that H3K4me1 and H3K27ac prime and activate enhancers, respectively. In contrast to this sequential action model, our data support a reciprocal cooperation model. Specifically, we show that CREBBP interacts with KMT2D, they are required for each other's stable chromatin association, and that losing either enzyme reduces the levels of both H3K4me1 and H3K27ac. This reciprocal cooperation model is plausible given how these histone marks are normally regulated during GC reaction. Specifically, previous reports showed that enhancers activated by CREBBP and KMT2D become transiently repressed in the DZ. This was due to actions of BCL6, a GC master regulatory transcriptional repressor. BCL6 binds mainly to enhancers, where it mediates H3K27 deacetylation through recruitment of HDAC3, and H3K4me1 demethylation through recruitment of KDM1A[17,49,50]. It is believed that BCL6 repressor complexes dissociate in the LZ, allowing CREBBP and KMT2D to rapidly and simultaneously restore enhancer activation marks[51,52], activities of which are likely further enhanced through immune synapse signals received during T cell help.

Given these considerations, our data suggest that the non-catalytic scaffold function of CREBBP and KMT2D plays an important role in their intimate cooperation, reminiscent of a prior study reporting that UTX bridges the interaction between P300 and KMT2D[53]. However, there are other potential ways that CREBBP and KMT2D could tune each other's activity. For example, it is possible that CREBBP-catalyzed H3K27ac weakens histone tail-DNA interactions to open chromatin structure, which in turn makes the H3K4 site accessible to KMT2D[54]. Conversely, KMT2D was recently appreciated to contain intrinsically disordered regions (IDRs) that confer its ability to undergo liquid-liquid phase separation[55]. Interaction of CREBBP with KMT2D could concentrate CREBBP into the phase separated transcriptional condensates, thereby boosting the enzymatic activity of CREBBP in depositing histone acetylation[56]. We also cannot rule out that some portion of the observed effects could be linked to putative roles of these proteins on modifying non-histone proteins. For example, a recent study also observed germinal center hyperplasia with CK double loss of function, and demonstrated that CREBBP forms a complex with KMT2D and directly acetylates KMT2D to modulate KMT2D activity at gene enhancers[57], further emphasizing the various facets of cooperation between CREBBP and KMT2D in GC B cells. Teasing apart the relative contribution of these possible mechanisms will require an extensive biochemical effort and will provide additional important basic insight into the effects we observed herein.

It is notable that super-enhancers were especially vulnerable to depletion of both CREBBP and KMT2D, which may offer a key explanation for why their mutations are co-occurrent. Super-enhancers are complex regulatory elements composed of multiple clustered enhancers, often quite distal from the genes they regulate, and have been described as regulating cell fate-determining genes[46]. Although some

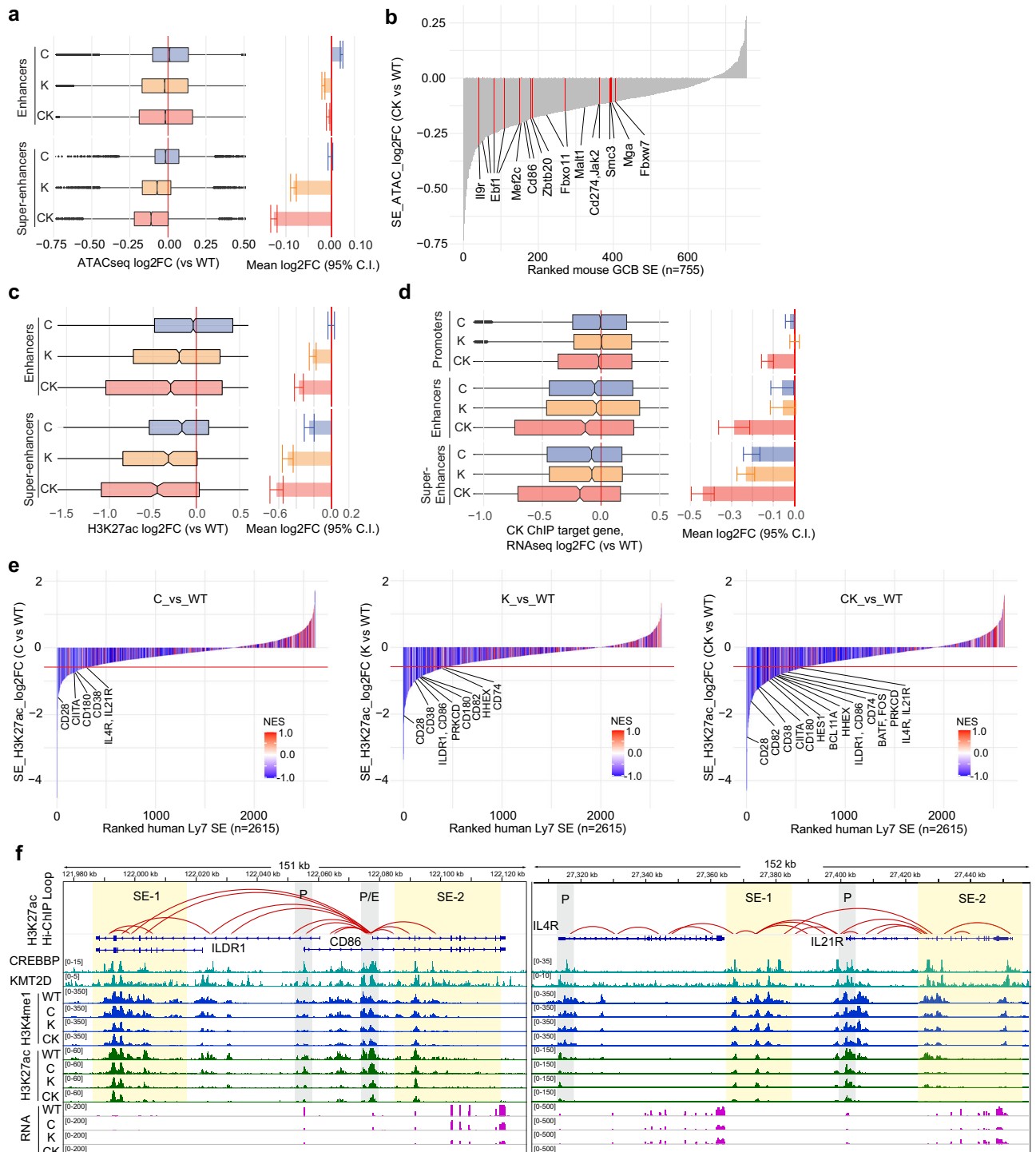

**Fig. 7 | CREBBP and KMT2D haploinsufficiency cooperatively impairs activation of super-enhancers controlling GC B cell fate decisions. a, c, d** Left: box plots showing C/K/CK deficiency-induced changes in chromatin accessibility in mouse GC B cells (**a**, enhancers: $n = 33,448$, super-enhancers: $n = 3746$), or H3K27ac in OCI-Ly7 cells (**c**, enhancers: $n = 2737$, super-enhancers: $n = 616$) at enhancer and constituent super-enhancer peaks, or changes in the target gene expression of CK-co-bound promoters, enhancers, and super-enhancers in OCI-Ly7 cells (**d**, $n = 5411$, 1959, 2031 genes respectively). The middle line in the box marks median. The box vertical size denotes IQR. The upper and lower hinges correspond to 25th and 75th percentiles. The upper and lower whiskers extend to the maximum and minimum values that are within 1.5 × IQR from the hinges. Right: bar plots showing the mean log2FC of chromatin accessibility (**a**), H3K27ac (**c**), or RNA (**d**), with error bars indicating 95% confidence intervals (C.I.) of the mean. **b** Waterfall plot ranking

super-enhancers (SE) in mouse GC B cells based on their accessibility change in CK vs WT. Constituent ATAC-seq peaks in each SE were summed before calculating the fold change. Genes linked to red-highlighted closing SEs were downregulated in CK vs WT. **e** Waterfall plots ranking super-enhancers in OCI-Ly7 cells based on their H3K27ac changes in C (left), K (middle) or CK (right) vs WT. Constituent H3K27ac peaks in each SE were summed before calculating the fold change. Bar colors represent NES values of RNA-seq GSEA, using all genes in each SE-residing TAD as gene signature against ranked gene list based on their expression changes in C/K/CK vs WT. Highlighted genes were downregulated and linked to the corresponding SEs. Red line indicates log2FC cut off of −0.58. **f** IGV views of normalized histone marks CUT&RUN and RNA-seq signals at the indicated loci in isogenic OCI-Ly7 cells. H3K27ac Hi-ChIP loop calls are depicted as arcs connecting the two interacting loci. Promoters and super-enhancers are shaded in gray and yellow colors, respectively.

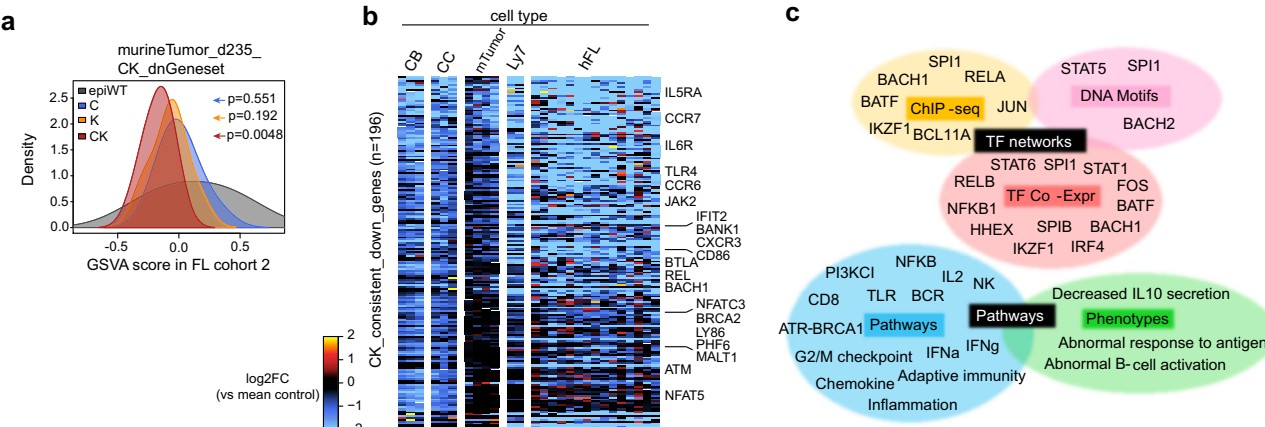

**Fig. 8 | CREBBP and KMT2D deficiency suppresses a core immune signature in GC B cells that is retained and shared between murine and human lymphomas. a** GSVA analysis using genes downregulated in BCL2 + CK vs BCL2 murine lymphoma cells at day 235 as gene set against human FL RNA-seq datasets (epiWT/C/K/CK: *n* = 16/10/8/15 patients). The p values were calculated using two-tailed Wilcoxon rank sum test. **b** Heatmap showing the relative expression levels of CK_consistent_down_genes (*n* = 196), including genes exhibiting downregulation in at least two of the indicated RNA-seq datasets. **c** Functional annotation of CK_consistent_down_genes by Enrichr and Toppgene.

reports claim that clustered enhancers can act in a synergistic manner, it is clear that super-enhancers are more additive in nature. This aligns with our finding of additive loss of super-enhancer function induced by loss of *CREBBP* and *KMT2D* alleles. Such additive super-enhancer effect may also be inferred by the additive cell fate decision defect observed, whereby intra-GC cell state transitions were progressively skewed to favor DZ retention over GC exit from C to K to CK mice.

It is intriguing that many of these CK dependent super-enhancers control immune synapse signaling genes that mediate cross talk with T cells, and direct B cell fate upon T cell help. The hallmark of GCB-DLBCL and FL is persistence of the GC transcriptional program that includes BCL6-mediated transient downregulation of many immune receptors that become re-expressed in the LZ. Failure to re-activate these genes is thus central to the phenotype of these tumors and may explain why *CREBBP* and *KMT2D* mutations are so highly prevalent, given that we show how strongly they impair these programs. Indeed CK deficiency impaired gene activation by most of the canonical LZ TFs including SPI1, BATF, NFkB, STAT3, AP1, IKZF1 and others[40,41]. This manifested as impaired expression of key genes involved in GC B cell interaction with TFH cells and GC exit. Notably among these was robust silencing of MHC class II, CD86, an important T cell costimulatory ligand that binds to CD28[58], as well as other key co-stimulatory genes such as CD40 and SLAMF1[59,60], and GC regulating chemokine receptors such as CCR6[61], the absence of which enhances GC formation and impairs affinity maturation. Collectively it might be anticipated that the sum of these perturbations would lead to reduction in immune tone and hence impaired activation and recruitment of T cells.

Along these lines, our data suggest that the interface of CK lymphoma cells with the immune microenvironment evolves during disease progression. At earlier stages there was a relative expansion of TFH cells, upon which CK GCB cells were especially dependent for their fitness. It is notable that this TFH cell expansion was out of proportion to TFR cells, which would further favor persistence of GC microenvironments. CD8+ cells are not normally part of the GC reaction and are more likely to mount an anti-tumor cytotoxic effect. Our data suggest that as they evolve, CK lymphomas may employ several integrated strategies to reduce CD8+ abundance and cytotoxic attack, including Treg expansion (Supplementary Fig. 2n, o), chronic suboptimal CD8+ stimulation (Fig. 6o–q) and enhanced CD8+ exhaustion (Fig. 1j). Finally, it might seem contradictory that CK deficient OCI-Ly7 cells were defective in activating human CD8+ cells when co-cultured at equal ratios, yet there were greater numbers of activated CD8+ cells in

CK mutant murine lymphomas. One possible explanation is that despite their defective stimulation at a 1:1 ratio, the greater expansion and local abundance of CK mutant lymphoma cells (in comparison to the other genotypes) provided more opportunity for suboptimal CD8+ activation, which has been reported to compromise the proliferation and effector functions of CD8+ cells[62].

Overall, the impaired ability of CD8+ cells to engage with CK deficient lymphoma cells may represent a form of aberrant super-enhancer mediated immune evasion, which likely contribute to accelerated pathogenesis, and have implications for improving the efficacy of T cell directed immunotherapies. Along these lines, it was shown that HDAC3 selective inhibitors given to counteract HDAC3 antagonism of CREBBP, recruited T cells into murine GCB-DLBCLs in vivo and enhanced the anti-tumor efficacy of checkpoint blockade[63]. Moreover, CREBBP mutant lymphoma cells were also more sensitive to HDAC3i in a cell autonomous manner, which has led to the concept of using these drugs for precision therapy in FL and DLBCL. Additionally, it was recently shown that KMT2D mutation sensitizes lymphoma cells to KDM5 inhibitors[64]. KDM5 histone demethylases remove H3K4me3 rather than H3K4me1, hence these drugs were thought to mediate their effects by impairing dynamic changes in H3K4 methylation status. We would predict that CK lymphomas would be highly sensitive to these agents given their cooperative actions, and perhaps even more to their combination. Hence our findings provide a basis for development of epigenetic combinatorial therapy with these agents, to reverse the effects of these mutations and induce both cell autonomous and immune anti-tumor activity. As these drugs become available, we propose they be tested in this context and used as immune adjuvant therapy together with T cell enhancing immunotherapies.

## Methods
### Ethics statement
Animal care was in strict compliance with institutional guidelines established by the Weill Cornell Medical College, the Guide for the Care and Use of Laboratory Animals (National Academy of Sciences 1996), and the Association for Assessment and Accreditation of Laboratory Animal Care International. FL patient specimens used for CD8 IHC staining (Fig. 1m, n) were previously published[35], acquired from the BC Cancer Agency lymphoma tumor bank and approved by the research ethics board of the University of British Columbia–British Columbia Cancer Agency (H13-01765).

## Mouse models

The following strains were obtained from The Jackson Laboratory (Bar Harbor, ME, USA): C57BL/6J (CD45.2/2, stock 000664), B6.SJL-Ptprca Pepcb/BoyJ (CD45.1/1, stock 002014), Cg1-Cre (stock 010611), Crebbp^flox (stock 025178). Kmt2d^flox (RRID:IMSR_JAX:032152) mice were obtained from Kai Ge at NIDDK[29]. The VavP-BCL2 (RRID:MGI:3842939)[31] model was developed by J.M. Adams (Walter and Eliza Hall Institute of Medical Research, Australia). Given there is no observed gender bias related to CREBBP and KMT2D mutations in human lymphomas, we didn't consider mice gender in experiment design. Mice were housed in solid-bottom, polysulfone, individually ventilated cages (IVCs) on autoclaved aspen-chip bedding, γ-irradiated feed, and acidified reverse osmosis water (pH 2.5 to 2.8) provided ad libitum. The IVC system is ventilated at approximately 30 air changes hourly with HEPA-filtered room air. The animal holding room is maintained at $72 \pm 2\,°F$ ($21.5 \pm 1\,°C$), relative humidity between 30% and 70%, and a 12:12 h light:dark photoperiod.

## Cell lines

The human GCB-DLBCL cell line OCI-Ly7 (RRID:CVCL_1881) was obtained from Ontario Cancer Institute and cultured in IMDM (ThermoFisher; 12440061) supplemented with 15% FBS, 1% L-Glutamine and 1% penicillin-streptomycin. The human GCB-DLBCL cell line SU-DHL4 (RRID:CVCL_0539) was obtained from DSMZ and grown in RPMI 1640 (ThermoFisher; 11875093) with 10% FBS, 1% L-Glutamine and 1% penicillin-streptomycin. Cell lines were authenticated by Biosynthesis using their STR Profiling and Comparison Analysis Service, and are routinely tested for mycoplasma contamination in the laboratory.

## Germinal center (GC) assessment in mice

To induce GC formation, age- and sex-matched mice were immunized intraperitoneally at 8 to 12 weeks of age with either 0.5 ml of 1:10 diluted sheep red blood cell (SRBC) suspension in PBS (Cocalico Biologicals) or 100 ug of the highly substituted hapten NP ($NP_{16}$ to $NP_{32}$) conjugated to the carrier protein ovalbumin (OVA) (Biosearch Technologies; N-5051) absorbed to Imject Alum Adjuvant (ThermoFisher; 77161) at a 1:1 volume ratio.

S-phase percentage of GC B cells was determined by EdU incorporation assay. Specifically, each mouse received 1 mg EdU dissolved in PBS through retro-orbital injection 1 h before euthanasia. After surface marker staining, splenocytes were subjected to fixation, permeabilization, and click reaction to fluorescently label incorporated EdU with AF488 following the instruction from the Click-iT Plus EdU Flow Cytometry Assay Kit (ThermoFisher; C10632).

For CD40L blocking assay, SRBC-immunized BM chimeric mice were administered with either anti-CD40L blocking antibody (Bio X Cell BE0017-1, 100 ug/mouse via IV) or IgG control antibody (Bio X Cell BE0091, 100 ug/mouse via IV) at day 4 and 6 post SRBC, followed by splenic GC analysis at day 8 post SRBC.

## Bone marrow transplantation and lymphomagenesis

Bone marrow (BM) cells were harvested from the tibia and femur of 8–12 years old donor mice. After red blood cell lysis using the ACK lysing buffer (Lonza; BP10-548E), $1 \times 10^6$ cells were injected into the retro-orbital sinus of lethally irradiated C57BL/6J recipient mice (2 doses of 450 rad, 12 h apart, on a Rad Source Technologies RS 2000 X-ray Irradiator). 8 weeks after transplantation to ensure full engraftment, mice were immunized with SRBC periodically (once every 3 weeks) to induce GC formation and the cells of origin for lymphomagenesis. Except for those euthanized at two specific time points (day 116 and day 235), all remaining mice were monitored until any one of several euthanizing criteria were met, including severe lethargy, more than 10% body weight loss, and palpable splenomegaly that extended across the midline, in accordance with Weill Cornell

Medicine Institutional Animal Care and Use Committee–approved animal protocol (protocol #2011-0031).

To generate mixed BM chimeric mice, B6.SJL mice were lethally irradiated with two doses of 450 rad X-ray 12 h apart and then transplanted with $1 \times 10^6$ mixed BM cells containing equal ratio of WT (CD45.1/2) and CK (CD45.2/2) through retro-orbital injection.

## Flow cytometry analysis and cell sorting

Single-cell suspension of mononuclear mouse splenocytes was prepared by Ficoll density gradient centrifugation (RD; I40650) and filtering through 35 μm nylon mesh. Cells were sequentially incubated with Zombie NIR (Biolegend 423106) in PBS (15 min at RT) for dead cell exclusion, rat anti-mouse CD16/32 (BD 553141, 1:1000) in FACS buffer for Fc receptor blockade (10 min at 4 °C), and fluorochrome-conjugated anti-mouse antibody mix diluted in Brilliant Stain Buffer (BD; 563794) for surface marker staining (30 min at 4 °C). For intracellular staining, cells were fixed and permeabilized using the Foxp3 staining kit (ThermoFisher; 00-5523-00), followed by intracellular antibody incubation (O/N at 4 °C). Data were acquired on Cytek Aurora, BD FACSymphony A5, or BD FACS Canto II flow cytometer analyzers, and analyzed using FlowJo software. The anti-mouse antibodies were diluted as follows: BUV615 anti-CD45 (BD 752418, Clone I3/2.3, 1:200), PerCP-Cy5.5 anti-B220 (BD 552771, Clone RA3-6B2, 1:200), PE-Cy7 anti-B220 (BD 552772, Clone RA3-6B2, 1:200), AF594 anti-B220 (Biolegend 103254, Clone RA3-6B2, 1:200), BV786 anti-B220 (BD 563894, Clone RA3-6B2, 1:200), BUV563 anti-CD38 (BD 741271, Clone 90/CD38, 1:250), BUV395 anti-CD38 (BD 740245, Clone 90/CD38, 1:250), APC-Cy7 anti-CD38 (Biolegend 102728, clone 90, 1:200), BV421 anti-FAS/CD95 (BD 562633, Clone Jo2, 1:200), PE anti-FAS/CD95 (BD 554258, Clone Jo2, 1:200), PE-Cy7 anti-FAS/CD95 (BD 557653, Clone Jo2, 1:200), PE anti-CXCR4/CD184 (BD 561734, Clone 2B11/CXCR4, 1:125), PE-Cy7 anti-CD86 (BD 560582, Clone GL1, 1:200), PerCP-Cy5.5 anti-CD45.1 (ThermoFisher 45-0453-82, Clone A20, 1:200), AF647 anti-CD45.2 (Biolegend 109818, Clone 104, 1:200), BUV737 anti-CD138 (BD 564430, Clone 281-2, 1:400), BV510 anti-IgD (BD 563110, Clone 11-26 c.2a, 1:100), APC anti-CD4 (Biolegend 100412, Clone GK1.5, 1:200), PE anti-CD8 (Biolegend 100708, Clone 53-6.7, 1:200), BV650 anti-IgG1 (BD 740478, Clone A85-1, 1:250), BUV661 anti-IgM (BD 750660, Clone II/41, 1:250), PerCP-eF710 anti-PD1 (Invitrogen 46-9985-82, Clone J43, 1:125), Biotin anti-CXCR5 (BD 551960, Clone 2G8, 1:50), AF532 anti-FOXP3 (Invitrogen 58-5773-80, Clone FJK-16s, 1:200), BV605 anti-CD62L (Biolegend 104437, Clone MEL-14, 1:200), BUV395 anti-CD44 (BD 740215, Clone IM7, 1:200), PE-Cy7 anti-CCR7 (Biolegend 120124, Clone 4B12, 1:125), PE anti-TCF1 (BD 564217, Clone S33-966, 1:200), AF594 anti-TOX (CST 61824, Clone E6G5O, 1:250). The anti-human antibodies were diluted as follows: PE anti-hCD8 (Biolegend 980902, Clone SK1, 1:200), BUV805 anti-hCD45RA (BD 742020, Clone HI100, 1:200), BV510 anti-hCD62L (BD 563203, Clone DREG-56, 1:200), PE-Cy5 anti-hCD95 (BD 559773, Clone DX2, 1:200), BV650 anti-hTNFα (BD 563418, Clone MAb11, 1:200), APC anti-hIFNγ (BD 554702, Clone B27, 1:200).

To sort mouse GCB, CB and CC, GCB cells were first enriched from mouse splenocytes using the PNA MicroBead Kit (Miltenyi Biotec, 130-110-479), followed by antibody incubation and sorting on BD Aria II sorter with 5 lasers (355, 405, 488, 561, 640). DAPI was used to exclude dead cells.

## Immunohistochemistry, immunofluorescence and imaging of tissues

Mouse organs (i.e., spleen, liver, lung, and kidney) were fixed in either 10% neutral buffered formalin for 24–48 h at RT (for IHC) or 4% paraformaldehyde for 12 h at 4 °C (for IF), then transferred to 70% ethanol for storage before submitting to the core facility Laboratory of Comparative Pathology at Weill Cornell Medicine for paraffin embedding, sectioning and staining.

Briefly, five micron-sections cut using Microtome (Leica RM2255) were deparaffinized and heat antigen retrieved in 10 mM sodium citrate buffer (pH 6.0) for 30 min using a steamer. For IHC, following antigen retrieval, endogenous peroxidase activity was blocked in 3% H2O2-methanol for 15 min at room temperature. Indirect IHC was performed using anti-species-specific biotinylated secondary antibodies followed by the introduction of avidin–horseradish peroxidase or avidin–alkaline phosphatase and developed by Vector Blue or DAB color substrates (Vector Laboratories). Sections were counterstained with hematoxylin. The following antibodies were used: Rat anti-mouse B220 (BD 550286, Clone RA3-6B2), Rabbit anti-mouse CD4 (ab183685, Clone EPR19514), and Rabbit anti-mouse CD8 (ab217344, Clone EPR21769). For IF, the sections were stained with primary antibodies Rat Anti-Mouse B220 (BD 550286, Clone RA3-6B2, 1:100 dilution) and Rabbit Anti-Mouse Ki67 (CST 12202, Clone D3B5, 1:250 dilution) overnight at 4 °C, followed by incubation with secondary antibodies Donkey anti-Rat IgG-AF488 (Invitrogen A21208, 1:500 dilution) and Donkey anti-Rabbit IgG-AF594 (Invitrogen A21207, 1:500 dilution) at room temperature for 1 h. Autofluorescence in tissue sections due to aldehyde fixation was removed by treatment with the TrueVIEW Autofluorescence Quenching Kit for 4 min (Vector Laboratories; SP-8400-15). Sections were counterstained with DAPI (ThermoFisher 62247, 1:1000 dilution) for 5 min.

Slides were scanned using the All-in-One Fluorescence Microscope (KEYENCE, BZ-X810). Fiji software (ImageJ) was used to quantify spleen and GC area size.

CD8 immunohistochemical staining was also performed on 4um slides of tissue microarray (TMA) from published follicular lymphoma cohort[35] using Benchmark XT platform (Roche Diagnostics, USA). Intrafollicular relative percentage of CD8 positive cells among overall cellularity was scored as previously described[65].

## Somatic hypermutation analysis of JH4 intron

Genomic DNA of sorted GC B cells were isolated by Quick-DNA Microprep Kit (Zymo Research; D3020) according to the manufacturer's protocol. JH4 intron sequences were amplified from genomic DNA by PCR with Phusion High-Fidelity DNA polymerase (NEB; M0530S) using JH4 forward primer and JH4 reverse primer. PCR products were resolved by agarose gel electrophoresis and JH4 intron sequences with size of 1.2 kb were extracted by QIAquick Gel Extraction Kit (Qiagen; 28706×4). DNA of JH4 sequences was ligated into the pCR-Blunt II-TOPO vector (ThermoFisher; K280002) and transformed into One Shot Chemically Competent *E. coli* according to the manufacturer's instruction. Clones were sequenced by Sanger sequencing with JH4 sequencing primer. Primer sequences can be found in Supplementary Data 7.

## Mouse CapIG-seq and clonality analysis

We performed CapIG-seq[66], a hybrid-capture sequencing assay, to enrich and sequence the rearranged VDJ regions of the B-cell receptors (BCRs). Briefly, 250 ng of DNA per sample was sheared to 250 bp segments using the Covaris LE220 and then subject to DNA library construction using the KAPA HyperPrep kit (Roche, KK8502). We next followed the IDT xGen hybridization capture of DNA libraries protocol to capture the BCR VDJ fragments from the library using a custom probe set designed through the IDT xGen Hyb Panel design tool. The captured libraries were then sequenced at the Princess Margaret Genomics Centre (Toronto, Canada) using NovaSeq 6000 (PE150bp), to the depth of 10–30 million reads per sample.

The CapIG-seq raw data was processed following the MiXCR pipeline[67]. To evaluate the diversity of BCR repertoire, we used the iNEXT (version 3.0.0) R package[68] to generate the diversity (defined by the Shannon Diversity score) and clonal fraction plots. For BCR clonality analysis, we first computed the Simpson diversity index (D) using

iNEXT. We then computed the square root of the reciprocal of D ($\sqrt{1/D}$) to get the Productive Simpson Clonality score.

## CRISPR/Cas9-mediated gene editing

To generate CREBBP-KO, CREBBP-R1446C, KMT2D-KO, and double loss-of-function isogenic OCI-Ly7 cells, ribonucleoprotein (RNP) complex containing Alt-R recombinant S.p. HiFi Cas9 Nuclease (IDT; 1081061), Alt-R CRISPR-Cas9 tracrRNA (IDT; 1072534), and Alt-R CRISPR-Cas9 crRNA targeting CREBBP exon 26 or KMT2D exon 4, were assembled in vitro following the manufacturer's protocol and delivered into OCI-Ly7 cells by electroporation using SF Cell Line 96-well Nucleofector Kit (Lonza; V4SC-2096). ssODN for CREBBP R1446C mutagenesis was added along with RNP complex during electroporation as needed. Single cells were then seeded in 96-well plates by serial dilution. Clones were screened by Sanger sequencing of PCR amplicons and immunoblot. crRNA, ssODN and genotyping primer sequences can be found in Supplementary Data 7.

## B-T co-culture assay

OCI-Ly7 cells were pulsed with either 2 ug/ml CEF peptide (STEMCELL, 100-0675) or equal volume of DMSO vehicle control for 2 h at 37 °C incubator, followed by wash and 60 Gy irradiation. HLA-A matched human CD8+ T cells (HLA: A*01:01/A*01:01) were purified from cryopreserved human PBMC (STEMCELL, 70025.1) using the EasySep human CD8+ T cell isolation kit (STEMCELL, 17953). The purified CD8+ and irradiated OCI-Ly7 cells were suspended in complete RPMI1640 medium (supplemented with 10% FBS, 1XNEAA, 50 uM 2-Mercaptoethanol, 100 units/ml IL-2, 10 ng/ml IL-15) and then seeded at equal ratio in 96-well plate, followed by co-culture for 10 days (medium was refreshed once every two days). At day 10, cells were split into halves. One half was directly stained for surface markers, while the other half was re-stimulated with a cell stimulation cocktail (plus protein transport inhibitors) (Invitrogen, 00-4975) for 5 h at 37 °C incubator, followed by surface and intra-cellular cytokine staining.

## Co-immunoprecipitation and Immunoblotting

Nuclear extracts were prepared as previously described[69]. Nuclear extracts with salt adjustment (150 mM KCl) and detergent supplement (0.1% IGEPAL CA-630) were incubated with primary antibodies (CREBBP: Santa Cruz SC-369; KMT2D: MilliporeSigma ABE1867) overnight followed by addition of DynaBeads protein A (ThermoFisher; 10002D) the next day for 1.5 h. After extensive washes for 6 times, the beads were boiled in 1X Laemmli sample buffer, separated by SDS-PAGE, and analyzed by immunoblot using the indicated primary antibodies.

For immunoblotting, cell lysates were resolved by SDS−PAGE, transferred to PVDF membrane, and probed with the following primary antibodies: anti-CREBBP (Santa Cruz SC-369, 1:1000) anti-KMT2D (MilliporeSigma ABE1867, 1:1000), anti-MED1 (ab64965, 1:1000), anti-H3K4me1 (ab8895, 1:1000), anti-H3K27ac (ab4729, 1:1000) and anti-H3 (ab18521, 1:1000). Membranes were then incubated with a peroxidase-conjugated correspondent secondary antibody and detected using enhanced chemiluminescence. Densitometry values were obtained using Fiji software (NIH).

## ELISA and ELISPOT

Mice were immunized with 100 ug NP19-OVA in alum via intraperitoneal injection. Serum collected at different days after immunization was subjected to ELISA. Briefly, serum was incubated in 96-well plates coated with NP28-BSA or NP8-BSA to capture low and high affinity NP-specific antibodies, respectively, followed by quantification of IgG1 abundance using the SBA Clonotyping System-HRP kit (SouthernBiotech; 5300-05) following the manufacturer's instruction. For ELISPOT assay, bone marrow cells were collected on day 85 after immunization. 3 million BM cells were seeded into a 96-well

MultiScreen IP Filter Plate (MilliporeSigma; MSIPS4510) coated with NP8-BSA and incubated overnight at 37 °C. NP-specific LLPC spots were visualized using AP-conjugated goat anti-mouse IgG1 (SouthernBiotech; 1071-04) with enzyme substrates.

## Single-cell RNA-seq

Splenic B cells from mice immunized with SRBC for 10 days were pre-enriched by B220 MicroBeads (Miltenyi Biotec; 130-049-501) followed by flow sorting. Sorted B220+IgD- B cells from each spleen were submitted to the Epigenomics Core at Weill Cornell Medicine for library preparation and sequencing. Single-cell RNA-seq libraries were prepared using Chromium Single Cell 3' Reagent Kit according to 10x Genomics specifications. Four independent single-cell suspensions (one per genotype) with a viability of 70-80% and a concentration of 400-800 cells/ul, were loaded onto the 10x Genomics Chromium platform to generate barcoded single-cell GEMs, targeting about 8000 single cells per sample. GEM-Reverse Transcription (53 °C for 45 min, 85 °C for 5 min; held at 4 °C) was performed in a C1000 Touch Thermal Cycler with 96-Deep Well Reaction Module (Bio-Rad). After RT reaction, GEMs were broken up and the single-strand cDNAs were cleaned up with DynaBeads MyOne Silane Beads (ThermoFisher; 37002D). The cDNAs were amplified for 12 cycles (98 °C for 3 min; 98 °C for 15 s, 63 °C for 20 s, 72 °C for 1 min). Quality of the cDNAs was assessed using Agilent Bioanalyzer 2100, obtaining an average product size of 1815 bp. These cDNAs were enzymatically fragmented, end repaired, A-tailed, subjected to a double-sided size selection with SPRIselect beads (Beckman Coulter) and ligated to adaptors provided in the kit. A unique sample index for each library was introduced through 14 cycles of PCR amplification using the indexes provided in the kit (98 °C for 45 s; 98 °C for 20 s, 54 °C for 30 s, and 72 °C for 20 s x 14 cycles; 72 °C for 1 min; held at 4 °C). Indexed libraries were subjected to a second double-sided size selection, and libraries were then quantified using Qubit (ThermoFisher). The quality was assessed on an Agilent Bioanalyzer 2100, obtaining an average library size of 455 bp.

Libraries were diluted to 2 nM and clustered on an Illumina NovaSeq 6000 on a pair-end read flow cell and sequenced for 28 cycles on R1 (10x barcode and the UMIs), followed by 8 cycles of I7 Index (sample Index), and 98 bases on R2 (transcript), with a coverage around 200 M reads per sample. Primary processing of sequencing images was done using Illumina's Real Time Analysis software (RTA).

## Single-cell RNA-seq analysis

Fastq files were processed with cellranger version 3.0.0. This data was combined with previously published single cell datasets (H1, Ezh2 and Smc3) and processed using Seurat version 4.0[70]. Cell types of these integrated datasets were assigned together from previous annotations using the Seurat TransferLabel function and previous reference datasets[37]. Proper assignment of clusters was assessed by both individual genes, and previously published gene signatures. Gene signature module scores were calculated using the AddModuleScore function, with a control value of 5. Cells from this experiment were then subsetted and reprocessed using 5000 variable genes to generate the UMAP with the appropriate cell populations. Cell type percentages were calculated after removal of non-GC cells and significance was tested using a fisher's exact test. Pseudotime trajectories were generated with slingshot 2.4.0[38] on GC B cells. Density of cells along this pseudotime was plotted, and significance was tested using wilcoxon rank sum. Gene signatures were plotted against pseudotime and broken up into 10 deciles. Expression of the gene signatures in each of these deciles were tested using wilcoxon rank sum between the WT and mutant cells. The differences in the splines were plotted and colored according to decile significance.

## Bulk RNA-seq

Total RNAs were extracted from sorted cell suspensions using TRIzol LS Reagent (Invitrogen,10296010) following the manufacturer's instructions. RNA concentration was determined using the Qubit RNA High Sensitivity Kit (ThermoFisher, Q32855) and integrity was assessed using RNA 6000 Pico Kit (Agilent, 5067-1513) run on the Agilent 2100 Bioanalyzer. 100ng-400ng samples with RNA Integrity Number (RIN) > 9 were submitted to MedGenome for library preparation using the Illumina TruSeq Stranded mRNA Library Kit and PE100 sequencing on NovaSeq.

## Bulk RNA-seq analysis

Fastq files were processed with the nfcore RNA-seq pipeline version 2.0, aligned to mm10 or hg38. Count files from the nfcore pipeline output were processed with DESeq2. PCA clustering was performed on VST expression values for all GENCODE vM25 genes. Hierarchical clustering was then performed on the top 10% most variable genes using Euclidean distance and Ward's method. DESeq2 was also used to determine DEGs. Transcripts with less than 10 combined reads among all replicates or that DESeq2 failed to generate a p value for were removed from the datasets. P values were readjusted using Benjamini-Hochberg correction and differentially expressed genes were defined using a fold change <1.5 fcutoff and padj <0.01. Enrichment of signatures was performed using iPAGE. Briefly, unsupervised pathway analysis was performed using information-theoretic pathway analysis approach as described in[71]. Briefly, pathways that are informative about non-overlapping gene groups were identified. Pathways annotations were used from the Biological Process annotations of the Gene Ontology database (http://www.geneontology.org) and signature categories from the Staudt Lab Signature database[72]. Only human-curated annotations were used from the Gene Ontology database and only pathways with 5 genes or more, and with 300 genes or less were evaluated. This pathway analysis estimates how informative each pathway is about the target gene groups, and applies a randomization-based statistical test to assess the significance of the highest information values. We use the default significance threshold of $p < 0.005$. We estimated the false discovery rate (FDR) by randomizing the input profiles iteratively on shuffled profiles with identical parameters and thresholds, finding that the FDR was always less than 5%. For each informative pathway, we determined the extent to which the pathway was over-represented in the target gene group, using the hypergeometric distribution, as described in[73]. Clustering to define modules was performed on standardized log2 fold-change values relative to WT CB cells using fuzzy c-means clustering with 8 clusters and fuzzifier parameter as selected by Schwämmle-Jensen method[74].

## Bulk ATAC-seq

Bulk ATAC-seq libraries were prepared from sorted cell suspensions following the Omni-ATAC protocol[75]. Briefly, 60,000 freshly sorted live cells were washed in cold PBS, lysed in 60 ul cold lysis buffer (10 mM Tris-HCl, pH 7.5, 10 mM NaCl, 3 mM MgCl2, 0.1% NP-40, 0.1% Tween-20, and 0.01% Digitonin) for 3 min on ice, followed by addition of 1 ml wash buffer (10 mM Tris-HCl, pH 7.5, 10 mM NaCl, 3 mM MgCl2, and 0.1% Tween-20) and centrifugation to pellet nuclei. Nuclei were resuspended in a transposition reaction mix prepared using the Illumina Tagment DNA Enzyme and Buffer Kit (Illumina, 20034197) and incubated at 37 °C for 30 min on thermomixer at 1000 rpm to allow for transposition-based DNA fragmentation and adapter ligation. Tagged DNA fragments were purified using the Clean & Concentrator-5 Kit (ZYMO, D4014), then subjected to PCR amplification and double-sided bead purification (to remove primer dimers and larger than 1000 bp fragments) using the AMPure XP beads (Beckman Coulter, A63881). Library size distribution and quality was measured using the High Sensitivity DNA Kit (Agilent, 5067-4626) run on the Agilent 2100

Bioanalyzer. Libraries were submitted to MedGenome for PE50 sequencing on NovaSeq.

## Bulk ATAC-seq analysis

Fastq files were processed with the nfcore ATAC-seq pipeline version 1.2.1 and aligned to mm10. Count files were processed the same way as the RNA-seq data to generate PCA plots and hierarchical clustering. Differential peaks were also defined the same way as the RNA-seq, with the Benjamini-Hochberg adjusted pvalue < 0.01 and with a log2(1.5) fold change cutoff. These differential peaks were grouped according to k-means clustering, using clustGap function of the cluster R package (v2.1.0; Maechler, M., Rousseeuw, P., Struyf, A., Hubert, M., & Hornik, K. (2019). cluster: Cluster Analysis Basics and Extensions.) to calculate the optimal number of clusters. Supervised clustering was then performed by merging clusters of similar patterning across three genotypes. A fisher's exact test was then used to determine the odds ratio of the genomic feature of these differential peaks, using distance to TSS (±5kb) to define promoters, and H3K27ac or H3K4me1 to define putative poised or active enhancer types. Super-enhancers were defined using the ROSE algorithm[76,77] on previously published H3K27ac Mint-ChIP data, excluding any peaks within 2500 bp of a protein coding RNA transcription start site (TSS).

For RNAseq GSEA, gene expression was linked to differential ATACseq peaks using GREAT regulatory regions by basal plus extension model[78]. For ATACseq GSEA, peaks were linked to differentially expressed genes by previously defined mm10 germinal center TADs (H1). Constituent peaks of super-enhancers were used to determine differential accessibility at enhancers or super-enhancers, and significance was tested with wilcoxon rank sum. For the ranked Super-Enhancer accessibility, constituent peaks (overlap with the H3K27Ac super-enhancer peaks of at least 1 base pair) were summed per super-enhancer, then DESeq2 was used to normalize these counts with the size factors from the total ATACseq to account for read depth.

Peak clusters were determined by taking the variance stabilized transformed (VST) normalized counts in these merged regions for Mint-ChIP signal in H3K27Ac, H3K4me1, H3K4me3, and the ATAC-seq data from the four genotypes. t-SNE plots of these data were generated, and k-means clustering was used to define clusters. The clusters were linked with genes using distance to the closest TSS. The distance to TSS and the expression of these genes were used as a read-out of the cluster results.

TF motif enrichment was calculated as previously described[48]. Mainly, JASPAR mammalian motifs were scanned for in the ATACseq peaks, and the log2 fold-change of the peak accessibility change was linearly modeled as $\log 2FC \sim \beta_0 + x_1\beta_1 + \ldots + x_n\beta_n + GC*\beta_{GC}$, where $x_i$ denotes a boolean whether the $i^{th}$ transcription factor is present in the peak or not, and GC is the GC content of the peak. The effect sizes and p-values for each term are taken after multivariate linear modeling using ordinary least squares regression with robust standard errors.

## CUT&RUN

OCI-Ly7 cells were subjected to CUT&RUN assay according to the EpiCypher CUTANA CUT&RUN Protocol[79]. In brief, 500 K cells were immobilized onto activated ConA magnetic beads (Bangs Laboratories, BP531), followed by permeabilization and incubation with 0.5 ug of antibodies (H3K4me1, 13-0040; H3K4me3, 13-0028; or H3K27ac, 13-0045; SNAP-Certified for CUT&RUN and ChIP by EpiCypher) overnight at 4 °C. On the next day, the cell-bead slurry was washed and incubated with pAG-MNase (1:20 dilution, EpiCypher, 15-1116) for 10 min at RT. MNase was then activated by addition of CaCl2 to cleave targeted chromatin for 2 h at 4 °C. After chromatin digestion, MNase activity was stopped and chromatin fragments released into supernatant were purified using the NEB Monarch DNA Cleanup Kit (NEB, T1030) per manufacturer's instruction. 10 ng DNA was subjected to library preparation using NEBNext Ultra II DNA Library Prep Kit for Illumina (NEB, E7645) according to the manufacturer's protocol. Libraries were loaded onto the Illumina HiSeq for pair-end 150 bp sequencing with a sequencing depth of around 6 M reads per sample.

## Mint-ChIP sequencing

Mint-ChIP was performed using anti-H3K4me1 (Abcam ab8895) and anti-H3K4me3 (Abcam ab8580) as previously described[80]. Briefly, 5 × 10⁴ flow-sorted mouse GC B cells in triplicate were processed with MNase to fragment native chromatin. Barcoded adaptors were ligated to every sample, and samples were multiplexed for ChIP. After ChIP, material was linearly amplified by RNA in vitro transcription in the presence of the RNase inhibitor RNaseOUT (Invitrogen, 10777019). Fragments were reverse transcribed and amplified as a library. Sequencing was run on Illumina NextSeq 500.

## ChIP-seq and CUT&RUN analysis

Previously published CREBBP and KMT2D ChIP-seq data was aligned to hg38, with peaks being called with MACS2. The number of overlapped ChIP-seq peaks were calculated using the bedtools intersect function. Cut&Run fastq files were processed with nfcore cutandrun version 2.0, with IGG controls used per genotype. Read depth was normalized to reads at promoters. Peaks were called with SEACR and summits were called with MACS2. Counts were processed with DiffBind 3.6.1[81]. The union of consensus peaks from all marks was taken. Reads under the merged consensus peak was calculated for each mark by SubRead featureCounts[82] and normalized by variance stabilized transform (VST)[83]. t-SNE was generated using the Cut&Run signal from all histone marks for baseline WT samples and additional features of log2 fold-changes of each genotype compared to WT. The regions were clustered using k-means clustering. RNA expression was linked by taking the closest gene transcription start site (TSS) to each peak. Super-enhancers were defined using ROSE on the WT H3K27Ac Cut&Run data, with noncoding RNA TSS removed and default settings (2500 bp from TSS). Cut&Run heatmaps were generated with deepTools 3.5.1[84]. For H3K4me1, peak summits that overlapped with Cluster 1 peaks were used to generate peaks. For H3K27ac, peak center of peaks called by SEACR were used. RNA expression for both Cut&Run and CHIP was linked by GREAT and previously defined TADs.

## ChIP-qPCR

CREBBP and KMT2D ChIP assays were performed as previously described[17,18]. 50 million cells were first cross-linked with 2 mM DSG (disuccinimidyl glutarate, ThermoFisher, 20593) for 45 min at RT, and then 1% formaldehyde for 10 min at RT in PBS. Fixed cells were lysed to isolate nuclei, followed by sonication, incubation with Dynabeads (ThermoFisher, 11204D) coated with CREBBP (Santa Cruz, SC-369) or KMT2D antibody (Millipore, ABE1867), wash, elution, reverse-crosslink, and DNA purification. The recovered ChIP and input DNA was amplified by real-time quantitative PCR using SYBR Green (Applied Biosystems, 4385614) on QuantStudio 6 Flex Real-Time PCR System (Applied Biosystems). Data was analyzed by percent input method. ChIP-qPCR primers were listed in Supplementary Data 7.

## RT-qPCR

cDNA was prepared from extracted RNA using cDNA synthesis kit (ThermoFisher Scientific, K1641) and detected by fast SYBR Green (Applied Biosystems, 4385614) on QuantStudio 6 Flex Real-Time PCR System (Applied Biosystems) using the primers listed in Supplementary Data 7. qPCR signal for each gene was normalized to those of Gapdh (mouse) or HPRT (human) using the ΔCT method. Results were represented as fold expression relative to WT with the standard deviation for 3-4 biological replicates.

## Statistics and reproducibility

Statistical methods used for p value calculation were specified in the corresponding figure legends. Statistical analyses were conducted using either GraphPad Prism 9 or the R statistical language scripts and packages specified. Data were judged to be statistically significant when p or adjusted $p < 0.05$ unless otherwise noted. All experiments were successfully repeated at least three times except for single-cell RNA-seq which was performed using one mouse per genotype, and OCI-Ly7 RNA-seq and Cut&Run, which were performed using two biological replicates per genotype. No statistical method was used to predetermine sample size, which was estimated based on previously published papers[17,18]. No data were excluded from the analyses. The experiments were not randomized except for the CD40L blocking assay. The data were analyzed by groups and the investigators were blinded to group allocation during experiments.

## Reporting summary

Further information on research design is available in the Nature Portfolio Reporting Summary linked to this article.

## Data availability

Genomics data generated in this study have been deposited in the NCBI Gene Expression Omnibus (GEO) database under super series accession number GSE224513 (WES: GSE260809, CapIG-seq: GSE260810, RNA-seq: GSE224465, ATAC-seq: GSE224512, Single-cell RNA-seq: GSE224782, Cut&Run: GSE224781, Mint-ChIP: GSE260811). We also used the following previously published datasets: CREBBP (GSE133102)[42] and KMT2D (GSE67314)[18] ChIP-seq, H3K27ac HiChIP (GSE183797)[48] in OCI-Ly7, H3K27ac Mint-ChIP in mouse GC B cells (GSE146753)[39]. Source data are provided with this paper.

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

## Acknowledgements

The authors thank Dr. Kai Ge (NIH) for providing the *Kmt2d* floxed mice used in this study. We thank Dr. Martin T. Wells (Cornell University) for advice on statistical analysis. We thank all the members in the Melnick lab for their helpful discussion and technical support. In addition, we thank Weill Cornell Medicine core facilities, including Epigenomics Core, Genomics Resources Core, Flow Cytometry Core, and Laboratory of Comparative Pathology for their professional support of this work. This work is supported by grants A.M.M.: R35 CA220499, LLS SCOR 7021-20, IFLI collaborative research award. W.B. is supported by NCI R01 CA270245, ASH Junior Faculty Scholar Award, Leukemia & Lymphoma Society, Lymphoma Research Foundation, and The Follicular Lymphoma Foundation. C.R.C. is supported by 5 F31 CA254302-02. H.-Y.Y. was supported by AACR-Takeda Oncology Fellowship in Lymphoma Research. M.A.R. is supported by the ASH Junior Faculty Scholar Award. This work is supported by program project grant funding from the Terry Fox Research Institute (grant 1061) and the BC Cancer Foundation. D.W.S. is supported by a Michael Smith Foundation for Health Research Health Professional Investigator award (18646). R.G.R. was supported by grants R01AI148387 and LLS SCOR 7021-20. C.E.M. thanks the Scientific Computing Unit (SCU) at WCM, the WorldQuant Foundation, the National Institutes of Health (R01MH117406, R01CA249054, P01CA214274), and the LLS (MCL7001-18, LLS-9238-16, LLS-MCL7001-18, LLS 7029-23). O.W. is supported by the Else Kröner Excellence Fellowship (Else Kröner-Frese-nius-Stiftung, 2021_EKES.13), the Lymphoma Research Foundation (Jaime Peykoff Follicular Lymphoma Initiative), and German Research Foundation (DFG, WE 4679/2-1) and the Wilhelm Sander-Stiftung (2022.093.1 to O.W.).

## Author contributions

Conceptualization, J.L., C.R.C., H.Y.Y, W.B., and A.M.M.; formal analysis, J.L., C.R.C., H.Y.Y., C.M., M.R.T., P.F.,. K.T., C.S., D.W.S., Y.J., J.E., T.J.P., V.P., O.W., A.C., and A.M.M.; funding acquisition, C.R.C., H.Y.Y., C.M., W.B., C.E.M., and A.M.M.; investigation, J.L., C.R.C., H.Y.Y., C.M., M.R.T., M.A.R., M.X., C.S.C., Z.T., P.F., K.T., C.S., D.W.S., Y.J., J.E., T.J.P., V.P., O.W., A.C., R.Z., and A.M.M.; resources, C.E.M., R.G.R., D.W.S., W.B., and A.M.M.; writing - original draft; J.L., C.R.C., and A.M.M.; writing - review and editing; J.L., C.R.C., H.Y.Y., C.M., M.R.T., M.A.R., M.X., C.S.C., Z.T., P.F., K.T., C.S., D.W.S., Y.J., J.E., T.J.P., V.P., O.W., A.C., R.Z., C.E.M., R.G.R., W.B., and A.M.M.

## Competing interests

A.M.M. has research funding from Janssen, Epizyme and Daiichi San-kyo. A.M.M. has consulted for Exo Therapeutics, Treeline Biosciences, Astra Zeneca, Epizyme. C.S. has performed consultancy for Seattle Genetics, AbbVie, and Bayer and has received research funding from Bristol Myers Squibb, Epizyme and Trillium Therapeutics Inc. D.W.S. has received honoria from Abbvie, AstraZeneca, Incyte and Janssen and research funding from Janssen and Roche. C.E.M. is a cofounder and board member for Biotia and Onegevity Health as well as an advisor or grantee for Abbvie, ArcBio, Daiichi Sankyo, DNA Genotek, Tempus Labs, and Whole Biome. O.W. has research funding from Incyte and serves in the advisory board of BeiGene. T.J.P. has provided consultation for AstraZeneca, Chrysalis Biomedical Advisors, Merck, and SAGA Diagnostics (compensated); and receives research support (institutional) from AstraZeneca and Roche/Genentech. T.J.P. is an inventor on patents of the CapIG-seq and CapTCR-seq methods held by the University Health Network. R.Z. is inventor on patent applications related to work on GITR, PD-1 and CTLA-4. R.Z. is scientific advisory board member of iTEOS Therapeutics, and receives grant support from AstraZeneca and Bristol Myers Squibb. The remaining authors declare no competing interests.

## Additional information

[1]Division of Hematology/Oncology, Department of Medicine, Weill Cornell Medicine, Cornell University, New York, NY, USA. [2]Department of Physiology and Biophysics, Weill Cornell Medicine, New York, NY, USA. [3]The HRH Prince Alwaleed Bin Talal Bin Abdulaziz Alsaud Institute for Computational Biomedicine, Weill Cornell Medicine, New York, NY, USA. [4]BC Cancer Centre for Lymphoid Cancer, Department of Pathology and Laboratorial Medicine, University of British Columbia, Vancouver, Canada. [5]Centre for Lymphoid Cancer, British Columbia Cancer, Vancouver, Canada. [6]The Laboratory of Biochemistry and Molecular Biology, The Rockefeller University, New York, NY, USA. [7]Princess Margaret Cancer Centre, University Health Network, Toronto, ON, Canada. [8]Department of Medical Biophysics, University of Toronto, Toronto, ON, Canada. [9]Department of Medicine III, Laboratory for Experimental Leukemia and Lymphoma Research (ELLF), Ludwig-Maximilians University (LMU) Hospital, Munich, Germany. [10]Ontario Institute for Cancer Research, Toronto, ON, Canada. [11]Department of

Pathology and Laboratory Medicine, Weill Cornell Medicine, New York, NY, USA. [12]BC Cancer Centre for Lymphoid Cancer, Department of Medicine, University of British Columbia, Vancouver, Canada. [13]The WorldQuant Initiative for Quantitative Prediction, Weill Cornell Medicine, New York, NY, USA. [14]The Feil Family Brain and Mind Research Institute, Weill Cornell Medicine, New York, NY, USA. [15]Immunology and Microbial Pathogenesis Program, Weill Cornell Graduate School of Medical Sciences, New York, NY, USA. [16]These authors contributed equally: Jie Li, Christopher R. Chin, Hsia-Yuan Ying.
✉e-mail: web2002@med.cornell.edu; amm2014@med.cornell.edu

