## [Peer Review File · Nature Communications]

Loss of CREBBP and KMT2D cooperate to accelerate lymphomagenesis and shape the lymphoma immune microenvironmentREVIEWER COMMENTS

Reviewer #1 (Remarks to the Author):

The paper presented by Li and colleagues describes the phenotypic changes induced by combined GC-specific haploinsufficiency of the histone/chromatin modifier genes *Crebbp* and *Kmt2d*. The project has a high clinical relevance as the vast majority of lymphoma patients harbor inactivating mutations in both genes.

Based on their in vivo systems (using conditional mouse mutants) the authors gain insights in B cell lymphoma pathogenesis and provide potential rationalities for the development of enhancer targeting approaches to treat lymphoma patients.

Overall, the presented data is convincing and the manuscript is well written. However, the following limitations apply:

Major concerns:

1) The laboratories of Riccardo Dalla-Favera and Laura Pasqualucci recently published their findings on combinatorial *Crebbp* and *Kmt2d* inactivation in GC B cells and human DLBCL (Vlasevska et al. PNAS 2023). The authors' discussion about the competing manuscript is missing.

2) The authors' lymphoma development studies (based on a *Bcl2* transgenic background) are unique to the current study and their findings are very interesting. A detailed characterization of the aberrant B cell population will be even more informative:

The authors define lymphoma development by the histological appearance of the expanded cells. In addition, BCR clonality studies (by VDJ sequencing approaches) in the expanded B cell pool will discriminate between polyclonal B cell expansion (typically evident in *Bcl2* transgenic mice) and mono-/oligoclonal tumor development. In representative tumors Li and colleagues might want to confirm Cre-mediated combined loss of *Crebbp* and *Kmt2d* (e.g. by PCR based detection of exon loss in lymphoma cells).

The transcriptional and (epi)genetic profiling of the mono/oligoclonal tumor clones detected in the various genotypes (and the comparison of these mouse findings with human FL data

sets) would be very interesting as “tertiary” hits and insights in lymphoma evolution could be identified.

Beside the spleen do the authors observe B cell expansion in other lymphoid sites (e.g. peripheral LN, mes LN, PP, BM)? In the absence of transgenic Bcl2 overexpression, do the (aged) animals suffer from tumor development induced by GC-specific haploinsufficiency of both epigenetic modifiers?

The survival curve implies a similar disease course in BCL2+K and BCL2+CK mice, whereas BCL2+K animals will reach a survival plateau. Are these animals lymphoma-free or do they suffer from less-aggressive lymphoma entities?

3) The lack of cytotoxic T cell infiltration represents a key feature in BCL2+CK tumors compared to other genotypes. In contrast, CD4+ cells that form the immunological synapse required for B cell selection during the GC reaction, seem to be unaffected, at least their quantity in the tumor samples is independent of the underlying genotype. A more detailed characterization of the T cell pools infiltrating the lymphoma area (e.g. activation status, exhaustion marker expression, clonal selection) would be helpful to better understand the mechanisms of immune evasion which will take place during lymphoma development and progression. Furthermore, the authors have the unique opportunity to study T – B cell interactions in their autochthonous lymphoma models and may add some functional assays.

4) Crebbp and Kmt2d co-localize in a complex and impact on its expression levels and function by post-translational modifications (see Vlasevska et al. PNAS 2023).

The authors might want to prove the conservation of this interplay in their mouse tumors, e.g by determining Crebbp (and its paralog p300?) and Kmt2d protein expression and their modifications in the tumors.

Minor concerns:

Please report the percentage of combined mutations affecting Crebbp and Kmt2d in the data set used for Suppl. Figure 1 a.

The data in Fig 6 were generated predominantly in a single cell line (OCI-Ly7). The authors might want to use their murine lymphoma samples to validate selected findings in another model system.

Have the authors acquired data for chronic GCs as detectable in mesLN (or PP)?

Is class switch recombination undisturbed upon Crebbp and Kmt2d haploinsufficiency? Impaired IgG1 switching might impact on the results in Suppl Fig 3 e-j.

Reviewer #2 (Remarks to the Author):

Li and colleagues conducted in vivo experiments to investigate the mechanisms of the concurrent mutations in CREBBP and KMT2D in both GC development and lymphomagenesis. They observed that the loss of CREBBP and KMT2D leads to a more severe lymphoma phenotype and unexpected immune evasion behavior in cancer cells. This loss results in reduced infiltration of CD8+ T cells and suppression of immune synapse genes, contributing to a weakened T-cell response and flawed cell fate decisions, possibly aiding cancer cells in eluding immune surveillance. Additionally, from an epigenetic standpoint, the loss of cooperation between C+K had a significant and profound effect on super enhancers, especially those responsible for regulating immune synapse signaling genes. Functional characterization of the mutations identified in lymphoma derived from humans is essential for a more comprehensive understanding of these genes' roles in lymphomagenesis. The manuscript is well-crafted, and some comments are listed below:

1. On page 6, lines 3-5, the authors calculated the p-values for the co-occurrence of CREBBP and KMT2D mutations in FL. However, the text does not specify the number of genes that were selected for this calculation. Moreover, it remains unclear whether the authors took into account the potential functional effects of the mutations, such as whether they were

damaging or not, in their analysis.

2. In Supplementary Fig. 1d, the authors concluded that BCL2+CK exhibited 'highly, larger, more' features compared to others. However, the conclusion lacks statistical support, leaving the basis for this assertion unclear.

3. On page 7, line 11, the authors noted that the SHM (somatic hypermutation) burden varied among different genotypes. However, they did not specify which isotype of VH was analyzed. Understanding whether the VH isotypes were class-switched (e.g., IgG) or not (e.g., IgM) would be relevant, as it could significantly influence the SHM levels.

4. In the case of human FL patients, the authors do not make it clear whether they observed any differences in prognosis among the C, K, CK, and WT patients. Likewise, the manuscript lacks information on whether similar conclusions were reached regarding CD8+ T cells in the different mutation groups of FL or DLBCL.

5. The authors concluded that the loss of function in CREBBP and KMT2D cooperates to accelerate FL development, manifesting more aggressive characteristics than either allele alone. However, Figure 1b and 1d do not visibly differentiate between BCL2+k and BCL2+CK. An explanation from the authors may be needed to clarify this apparent inconsistency.

6. It is unclear from the manuscript whether the authors examined the phenotype of CD8 or CD4 T cells in the BCL2+CK mice or human FLs, such as signs of exhaustion or hyperactivity

7. On page 7, the flow data reveals a similar abundance of CD4+ T cells but a reduction in CD8+ T cells in BCL2+CK mice. It would be crucial for the authors to specify which CD8+ cell subtypes were reduced, as this information could enhance the understanding of the immune response dynamics.

8. On page 16, line 2, the authors performed RNA sequencing on B220+ cells, identifying them as lymphoma cells. However, the manuscript does not specify the criteria used to define these cells as tumor cells.

9. On page 18, the authors mentioned impaired expression of key genes involved in GC B cell interaction with TFH cells and GC exit. However, how C+K differentially modulate the infiltration of CD8+ and CD4+ T cells remains unclear, given that BCL2+CK mice mainly had reduced CD8+ T cells rather than CD4+ T cells. Further clarification on this aspect is needed.

Reviewer #3 (Remarks to the Author):

In this interesting manuscript, Li and colleagues investigate the effects of combined heterozygous loss of CREBBP and KMT2D in lymphoma, using the VavP-Bcl2 model and the GC B-cell specific Cg1-Cre strain. Using a variety of in vitro and in vivo analyses, authors show that combined loss of CREBBP and KMT2D induces a more severe phenotype than loss of each of them separately. This is associated to a phenotype of immune evasion with drastically reduced infiltrating CD8 T-cells. Furthermore, authors show the cooperative nature of CREBBP and KMT2D in the regulation of their epigenetic target programs, and even demonstrate that CREBBP and KMT2D co-interact. Finally, the epigenetic phenotype upon combined KMT2D/CREBBP loss was especially strong at superenhancer regions (and, more specifically, at super enhancers driving the expression of immune synapse signaling genes). Overall, this is a very comprehensive study uncovering novel relevant biology, and I would like to congratulate authors for their really nice work.

My only minor comment to authors is that it would be good if they could show all 4 genotypes in Fig 4i (as of know, it is not clear if the anti-CD40 bar is grouping all 4 genotypes, or if it's just one genotype only). Similarly, it would be good if authors could show the other 2 genotypes in Fig 4j, beyond the WT and CK, so that we can evaluate potential differences between single and combined genotypes.

REVIEWER COMMENTS

Reviewer #1 (Remarks to the Author):

The paper presented by Li and colleagues describes the phenotypic changes induced by combined GC-specific haploinsufficiency of the histone/chromatin modifier genes *Crebbp* and *Kmt2d*. The project has a high clinical relevance as the vast majority of lymphoma patients harbor inactivating mutations in both genes.

Based on their in vivo systems (using conditional mouse mutants) the authors gain insights in B cell lymphoma pathogenesis and provide potential rationalities for the development of enhancer targeting approaches to treat lymphoma patients.

Overall, the presented data is convincing and the manuscript is well written. However, the following limitations apply:

Answer: We thank the reviewer # 1 for these positive comments on our work.

Major concerns:

R1.1. The laboratories of Riccardo Dalla-Favera and Laura Pasqualucci recently published their findings on combinatorial *Crebbp* and *Kmt2d* inactivation in GC B cells and human DLBCL (Vlasevska et al. PNAS 2023). **The authors' discussion about the competing manuscript is missing.**

Answer: We thank the reviewer for this comment. We have added the following text to the discussion (all additions to the manuscript are in blue text):

“a recent study also observed germinal center hyperplasia with CK double loss of function, and demonstrated that CREBBP forms a complex with KMT2D and directly acetylates KMT2D to modulate KMT2D activity at gene enhancers¹, further emphasizing the various facets of cooperation between CREBBP and KMT2D in GC B-cells.”

R1.2. The authors' lymphoma development studies (based on a *Bcl2* transgenic background) are unique to the current study and their findings are very interesting. A detailed characterization of the aberrant B cell population will be even more informative:

The authors define lymphoma development by the histological appearance of the expanded cells. In addition, **BCR clonality studies (by VDJ sequencing approaches) in the expanded B cell pool will discriminate** between polyclonal B cell expansion (typically evident in *Bcl2* transgenic mice) and mono-/oligoclonal tumor development. In representative tumors Li and colleagues might want to **confirm Cre-mediated combined loss of *Crebbp* and *Kmt2d* (e.g. by PCR based detection of exon loss in lymphoma cells).**

Answer: We thank the reviewer for these suggestions.

1) Regarding the first of these two points, we performed Ig heavy chain VDJ region sequencing to better assess lymphoma clonal architecture. Overall, *BCL2*+CK lymphomas tend to comprise fewer BCR-distinct clones than *BCL2* (**Revision Fig. 1a**). A similar trend of clone number reduction was observed in *BCL2*+C and *BCL2*+K, although to a lesser degree. Consistently, a

more quantitative measure of BCR clonality by calculating either Simpson clonality score or Shannon diversity score revealed that BCL2+CK lymphomas tend to be more clonal and accordingly less diverse than BCL2, with BCL2+C and BCL2+K adopting an intermediate phenotype (**Revision Fig. 1b-c**). We included this new data in the revised manuscript as **Supplementary Fig. 2a-b**.

Revision Fig. 1. BCR immuno-seq (targeting Ig heavy chain VDJ regions) of day 235 murine lymphoma samples. **a**, Bar graphs depicting the number and fraction of all distinct IgH clones in each genotype. Each bar represents 1 mouse, 4 mouse replicates per genotype. **b-c**, Pie charts showing the fraction distribution of different IgH clonality (**b**) or diversity (**c**) categories in each genotype. The 25th and 75th percentiles were used as cut-off for intermediate vs low and high vs intermediate respectively.

2) Regarding the second part of the reviewer's question, we have confirmed the Cre-mediated heterozygous exon deletion of *Crebbp* and/or *Kmt2d* in day 235 murine lymphoma samples by genotyping PCR (**Revision Fig. 2**), which is also included as the new **Supplementary Fig. 1b-c**.

Revision Fig. 2. Genotyping PCR confirming the Cre-mediated heterozygous knock out of *Crebbp* (**a**) or *Kmt2d* (**b**) in day 235 murine lymphoma samples (4 replicates per genotype). Top: PCR primer location, with LoxP sites depicted by gray triangles. Bottom: agarose gel image of PCR product.

R1.3. The transcriptional and (epi)genetic profiling of the mono/oligoclonal tumor clones detected in the various genotypes (and the comparison of these mouse findings with human FL data sets) would be very interesting as **“tertiary” hits** and insights in lymphoma evolution could be identified.

Answer: We thank the reviewer for raising this interesting point. We have some experience performing exome sequencing on various lymphoma models and have not found these to be informative, since there are never enough numbers of distinct individual mice to show statistically significant patterns (these patterns are only evident when hundreds of human patients are profiled). Moreover, tertiary hits in mice have particularities that may be related to how lymphomas progress in the context of model systems with many competing clones. Regardless, in appreciation of the reviewer's interest in this topic we performed whole exome sequencing of day 235 murine lymphoma samples (4 replicates per genotype). After filtering out putative germline variants, defined as SNVs (single nucleotide variants) shared among two or more replicates, we obtained 24 highly enriched gene mutations (VAF>0.3) among all samples (**Revision Table 1**). As we anticipated, the biological impact of these mutations remains unclear and as such we did not include these data in the revised manuscript.

Gene	B_1	B_2	B_3	B_4	BC_1	BC_2	BC_3	BC_4	BK_1	BK_2	BK_3	BK_4	BCK_1	BCK_2	BCK_3	BCK_4
Mctp1	0.67															
E2f2		1.00														
Nvl*		0.60														
Ryr2		0.43														
Lzts3*		0.40														
Adgrf5			0.31													
Acyp2				0.31												
Wdtdc1					0.50											
Ahi1					0.50											
Plxnd1						0.36										
Lzts3*							0.40									
Dlec1							0.33									
Zfp180								0.41						0.30		
Ralgapa 2								0.32								
Dmbx1									0.31							
Cul4b										0.31						
Cpeb3											0.33					
Nvl*											0.30					
Atp13a3											0.30					
Igfbp7												0.67				
Pet2												0.43				
Nvl*													0.46			
Lrrfp1													0.40			
Slc25a47																0.44

Revision Table 1. A list of genes carrying unique, high-severity *de novo* mutations in day 235 murine lymphoma samples as revealed by whole exome sequencing. The values represent VAFs for the indicated mutant alleles (row names) in the indicated samples (column names). * denotes different mutations on the same gene.

R1.4. Beside the spleen do the authors observe B cell expansion in other lymphoid sites (e.g. peripheral LN, mes LN, PP, BM)?

Answer: We thank the reviewer for this question. To address this point we performed flow cytometry analysis of frozen bone marrow cells from our lymphoma mice cohort. As expected, the BCL2 allele resulted in increased abundance of B220+ B-cells across genotypes (**Revision Fig. 3a-b**), given the role of BCL2 in expanding B cell populations. Notably this effect was most evident at day 116 but not day 235 post BMT, for reasons that are currently unclear. Notably, the B220+ cell expansion in BCL2+CK was significantly higher than that in other genotypes (**Revision Fig. 3a-b**). To further understand these findings we next checked whether the observed B220+ cell expansion was derived from invading GCB-like cells, which more directly reflects the lymphoma phenotype. Using markers of normal GC B-cells (B220+CD38-FAS+) we observed enrichment for this population among all BCL2 containing genotypes as compared to WT (**Revision Fig. 3a, 3c**). However, lymphomas do not necessarily express identical markers as normal GC B-cells, and along these lines we observed a trend towards enrichment of GCB-like cells (B220+/FAS+) among the CK lymphomas vs the other BCL2 containing genotypes (**Revision Fig. 3a, 3d**). This result suggests that lymphoma B cells likely migrate to and expand in BM. However these findings require further study to better understand the biology of these cells. Apart from bone marrow, we unfortunately didn't collect LNs from this cohort so cannot show any data on those tissues.

Revision Fig. 3. Flow cytometry profiling of BM cells from VavPBCL2 mouse lymphoma cohort. **a**, Representative FACS plots show the gating strategy and frequency of total B (B220+), GCB (B220+CD38-FAS+), or GCB-like (B220+/FAS+) cells in BM from indicated mice. **b-d**, FACS analysis showing the relative

abundance of BM total B (**b**), GCB (**c**), or GCB-like cells (**d**) normalized to CD45+, B220+, or B220+ cells respectively at day 116 and 235 post-BMT (mean \pm SD). Each dot represents a mouse (n=4 mice per genotype). Statistical significance was determined using ordinary one-way ANOVA followed by Tukey-Kramer's multiple comparisons test (*p < 0.05; **p < 0.01; ***p < 0.001).

R1.5. In the absence of transgenic Bcl2 overexpression, do the (aged) animals suffer from tumor development induced by GC-specific haploinsufficiency of both epigenetic modifiers?

Answer: We thank the reviewer for bringing up this interesting point. We didn't include CK haploinsufficiency alone mice (without BCL2 overexpression) in our murine lymphoma cohort. Given that CK mutant human lymphomas virtually always carry BCL2 translocations, we reasoned that CK alone setting would not be as informative for studies of the human disease. Although it might be interesting conceptually, we decided to restrict our use of mice to those experiments that were most directly relevant to the physiological scenario, in part given concerns regarding excessive use of vertebrate organisms.

R1.6. The survival curve implies a similar disease course in BCL2+K and BCL2+CK mice, whereas BCL2+K animals will reach a survival plateau. Are these animals lymphoma-free or do they suffer from less-aggressive lymphoma entities?

Answer: We thank the reviewer for this interesting question. We didn't examine in further detail those few BCL2+K mice still alive at the end of our 300-day monitoring period. However, given that likely malignant GCB cell expansion was uniformly observed in randomly selected, healthy-looking BCL2+K mice at day 235 post BMT (**Fig.1d**), we speculate that those few late-surviving BCL2+K mice may have had lower grade lymphomas rather than being lymphoma-free.

R1.7. The lack of cytotoxic T cell infiltration represents a key feature in BCL2+CK tumors compared to other genotypes. In contrast, CD4+ cells that form the immunological synapse required for B cell selection during the GC reaction, seem to be unaffected, at least their quantity in the tumor samples is independent of the underlying genotype. A more detailed characterization of the T cell pools infiltrating the lymphoma area (e.g. activation status, exhaustion marker expression, clonal selection) would be helpful to better understand the mechanisms of immune evasion which will take place during lymphoma development and progression. Furthermore, the authors have the unique opportunity to study T – B cell interactions in their autochthonous lymphoma models and may add some functional assays.

Answer: We thank the reviewer for this excellent suggestion. *To address the first point* we performed Cytex multi-dimensional flow cytometry analysis of T-cell phenotypes (naive, central memory, effector, exhaustion, regulatory T cells, Tfh, Tfr) using the remaining frozen splenocyte specimens from our lymphoma cohort mice at both time points to gain more insights into how CK loss of function differentially affects CD4 and CD8 T cells in the lymphoma microenvironment. As noted before, CD8, but not CD4, displayed progressive reduction from BCL2+C, BCL2+K, to BCL2+CK, relative to BCL2 at both time points (**Revision Fig. 4a-b**). Given that all subsequent analyses were based on these specimens, we replaced the former flow cytometry analysis from **Fig. 1g-h** with these data, to directly match the more detailed flow cytometry results.

Revision Fig. 4. FACS analysis showing the relative abundance of splenic CD4+ (a) or CD8+ (b) cells normalized to CD45+ cells (mean \pm SD) in mice from VavPBCL2 lymphoma cohort. Each dot represents a mouse (n=4 mice per genotype). Statistical significance was determined using ordinary one-way ANOVA followed by Tukey-Kramer's multiple comparisons test (*p < 0.05; ***p < 0.001; ****p < 0.0001).

On the CD8 side, at day 116 we observed significant expansion of CD8 effector cells (CD8+CD44+CCR7-), accompanied by a corresponding reduction of central memory (CM, CD8+CD44+CCR7+) CD8 cells in BCL2+CK, that was greater than effects observed in BCL2+C and BCL2+K (**Revision Fig. 5a-b**).

We note for the reviewer that day 235 specimens were actually thawed and examined in our lab prior to the day 116 tumors mentioned just above. Unfortunately, we were unable to accurately separate naive, CM and effector cells at day 235 due to failure of the CD62L channel, caused by the known pitfall of shedding and loss of this marker (that we were initially unaware of) after freeze-thaw cycle as previously reported². This was why we switched to subsequent labeling of CCR7 in our day 116 samples.

This issue notwithstanding, at day 235 we noted significant expansion of CD44+ activated CD8 T cells (which contain both effector and CM populations) in BCL2+C and BCL2+CK, whereas the fraction of these cells in BCL2+K was reduced compared to BCL2 (**Revision Fig. 5c-d**). This interpretation is limited by the lack of CD62L staining at day 235 and so should not be viewed as conclusive.

Most remarkably, even though the abundance of CD8 cells was strongly reduced at day 235 in the BCL2+CK setting, **there was a massive and significant increase in the proportion of these cells manifesting an exhausted phenotype**, whereas the proportion of exhausted cells was roughly similar to BCL2 alone in the BCL2+C and BCL2+K setting (**Revision Fig. 5e-f**). In contrast there was no change in the abundance of exhausted CD8 cells at day 116. *Although we do not know the etiology of this phenotypic discovery, it does indicate significant CD8 T-cell dysfunction in the BCL2+CK setting. Lacking more specific information on the fraction of effector vs CM cells in this context we suggest it is best to avoid concluding at this point that the cause of exhaustion is linked to the apparent increase in activated CD8 T-cells. However, it does further underline how cooperative effects of C+K can alter the immune landscape and we thank the reviewer for encouraging us to add these interesting experiments to the manuscript.*

Revision Fig. 5. FACS-based immunophenotyping of splenic CD8 T cells from murine VavPBCL2 lymphoma cohort. **a, c, e**, Representative FACS plots show the gating strategy and frequency of different splenic CD8 subtypes (**a**, naïve/CM/effector CD8 at day 116, **c**, CD44+ activated CD8 at day 235, **e**, TCF1-TOX+ exhausted CD8 at day 116 and 235) in indicated mice. **b, d, f**, FACS analysis showing the relative abundance of splenic naïve/CM/effector CD8 at day 116 (**b**), CD44+ CD8 at day 235 (**d**), or TCF1-TOX+ exhausted CD8 at day 116 and 235 (**f**) normalized to CD8+ cells (mean \pm SD). Each dot represents a mouse (n=4 mice per genotype). Statistical significance in panels **b, d, f** was determined using ordinary one-way ANOVA followed by Tukey-Kramer's multiple comparisons test (*p < 0.05; **p < 0.01; ***p < 0.001; ****p < 0.0001).

On the CD4 side, TFH (T follicular helper, CD4+CXCR5+PD1+FOXP3-) cell frequency was progressively increased from BCL2+C, BCL2+K, to BCL2+CK, relative to BCL2 at both time points (**Revision Fig. 6a-c**). Intriguingly, TFH expansion occurred out of proportion with GC suppressive T follicular regulatory cells (TFRs, CD4+CXCR5+PD1+FOXP3+) at day 116 (**Revision Fig. 6d-e**). Given that TFRs are known to terminate the GC reaction in part by suppressing TFH functions^{3, 4}, we speculate that increased TFH/TFR ratios may favor initial expansion of malignant GC-like structures. Consistently, TFH/GCB ratios were significantly increased in BCL2+K and BCL2+CK compared to BCL2 at day 116 (**Revision Fig. 6f**). Finally, there was an equivalent increase in

Tregs (CD4+FOXP3+) among C/K/CK mutant genotypes relative to BCL2 in the end-stage lymphomas (**Revision Fig. 6g-h**), pointing to additional layers of immune dysfunction.

Revision Fig. 6. FACS-based immunophenotyping of splenic CD4 T cells from murine VavPBCL2 lymphoma cohort. **a-b**, Representative FACS plots show the gating strategy and frequency of splenic Tfh (CD4+CXCR5+PD1+FOXP3-) and Tfr (CD4+CXCR5+PD1+FOXP3+) cells in indicated mice at day 116 (**a**) or day 235 (**b**) post BMT. **c-d**, FACS analysis showing the relative abundance of splenic Tfh (**c**) or Tfr (**d**) normalized to CD4+ cells at day 116 and 235 post BMT (mean \pm SD). Each dot represents a mouse (n=4 mice per genotype). **e-f**, FACS analysis showing the ratio of Tfh vs Tfr (**e**) or Tfh vs GCB (**f**). **g**, Representative FACS plots show the gating strategy and frequency of splenic Treg (CD4+FOXP3+) in indicated mice at day 116 and 235 post BMT. **h**, FACS analysis showing the relative abundance of splenic Treg normalized to CD4+ cells at day 116 and 235 post BMT (mean \pm SD). Statistical significance in panels **c-f** and **h** was determined using ordinary one-way ANOVA followed by Tukey-Kramer's multiple comparisons test (* $p < 0.05$; ** $p < 0.01$; *** $p < 0.001$; **** $p < 0.0001$).

Together, these data suggest that CREBBP and KMT2D loss of function cooperatively remodels the immune microenvironment from a TFH-enriched pro-GC reaction early stage to a CD8-exhausted and depleted later stage to facilitate lymphomagenesis (**Revision Fig. 7**). These new CD4 and CD8 flow data are included in the revised manuscript as **Fig. 1g-j, 1o, Supplementary Fig. 2c-o**.

Lymphoma microenvironment change in BCL2+CK vs BCL2

Revision Fig. 7. A graphical model summarizing the lymphoma microenvironment change in BCL2+CK vs BCL2. Up and down arrows indicate population increase or decrease respectively in BCL2+CK vs BCL2. CD8cm: central memory CD8 (CD8+CD44+CCR7+), CD8eff: effector CD8 (CD8+CD44+CCR7-), CD8act: activated CD8 (CD8+CD44+, including both CM and effector populations), CD8ex: exhausted CD8 (CD8+TCF1-TOX+).

Regarding the reviewer's second point, given that we don't have ongoing lymphoma cohorts for functional studies of B-T interaction, as an alternative we performed in vitro co-culture assays by mixing irradiated isogenic OCI-Ly7 cells (WT and CK) pre-pulsed with either DMSO vehicle or CEF peptide (a pool of HLA class I-restricted virus peptides), with HLA-A matched human peripheral blood CD8 T cells at 1:1 ratio in the presence of IL2 and IL15 for 10 days, followed by FACS-mediated analysis of CD8 T cell activation and cytokine production (**Revision Fig. 8a**). We hypothesized that CK deficiency-induced reduction in immune synapse co-stimulatory proteins would result in weaker activation of CD8 cells.

As expected, CEF-pulsed OCI-Ly7 cells induced CD8 expansion to a greater degree than their DMSO-treated counterparts which presumably activated CD8 mainly through MHC-mismatch-triggered alloreactive response (**Revision Fig. 8b-d**). In both conditions, CK-deficient OCI-Ly7 cells were defective in activating and expanding CD8 T cells. However, it is noteworthy that this effect was more significant in the CEF treated setting, indicating a defect of CK cells in activating antigen-specific CD8 T-cell responses.

We next wondered whether this weaker stimulation results in defective T cell differentiation. Indeed, when zooming in to different CD8 subtypes, we observed an increase of naïve

(hCD45RA+hCD62L+hCD95-) and central memory (CM, hCD45RA-hCD62L+), and a concordant decrease of effector memory (EM, hCD45RA-hCD62L-) or effector (hCD45RA+hCD62L-) cells upon CK stimulation than WT (**Revision Fig. 8e-f**). Finally, re-stimulation of CK-exposed CD8 cells yielded reduced induction of IFN γ and TNF α (**Revision Fig. 8g-i**), two signature effector cytokines of cytotoxic CD8 cells involved in tumor clearance⁵, indicating that CK-stimulated CD8 cells were functionally less active.

These conditions allow dissection of isogenic WT vs CK T-cell stimulatory effects to be visualized at the single cell level with controlled ratios of cells and demonstrate impaired CD8 activation in vitro. This system enables observations to be made that cannot be dissected out from the syngeneic tumor microenvironment in vivo, thus providing a more complete picture of the immune perturbation induced by CK mutations, as requested by the reviewer. These new data are included in the revised manuscript as **Fig. 6o-q** and **Supplementary Fig. 6p-v**. The implications of our collective findings and further integration of these results are included in the **revised discussion section**.

Revision Fig. 8. In vitro B-T co-culture assay. **a**, Experimental design. **b-c, g**, Representative FACS plots show the gating strategy and frequency of different CD8 subtypes (**b-c**, naïve, CM, EM, and effector) or different cytokine-producing CD8 cells (**g**, DP, IFN γ +TNF α +, DN, IFN γ -TNF α -) in the indicated co-culture. **d-f, h-i**, FACS analysis showing the relative abundance of total CD8 (**d**, normalized to live cells), different CD8 subtypes (**e-f**, normalized to total CD8), or different cytokine-producing CD8 (**h-i**, normalized to total CD8) (mean \pm SD). Each dot represents one replicate (n=3 wells per co-culture). Statistical significance was determined by two-tailed unpaired student's t test.

R1.8. Crebbp and Kmt2d co-localize in a complex and impact on its expression levels and function by post-translational modifications (see Vlasevska et al. PNAS 2023). **The authors might want to prove the conservation of this interplay in their mouse tumors, e.g by determining Crebbp (and its paralog p300?) and Kmt2d protein expression and their modifications in the tumors.**

Answer: We thank the reviewer for this suggestion. Unfortunately, we are unable to test for protein modifications since we lack the special reagents optimized by Laura Pasqualucci over many years. Moreover, this point was already published in her paper and is not within the scope of our current manuscript which does not focus on the enzymatic biochemistry of these proteins relative to each other.

Minor concerns:

R1.9. Please report the percentage of combined mutations affecting Crebbp and Kmt2d in the dataset used for Suppl. Figure 1 a.

Answer: We thank the reviewer for this suggestion. We have included the percentage of double mutant patients in the revised **Supplementary Fig. 1a legend**. Specifically, 185 out of 478 patients (38.7%) in FL_all dataset and 37 out of 319 patients (11.6%) in EZB/Cluster 3 dataset carry CK double mutations.

R1.10. The data in Fig 6 were generated predominantly in a single cell line (OCI-Ly7). **The authors might want to use their murine lymphoma samples to validate selected findings in another model system.**

Answer: We thank the reviewer for this suggestion. Unfortunately, we did not have sufficient material frozen from our mouse lymphoma cohorts to perform assays as in **Fig.6**. Importantly however, our mouse GCB RNA-seq (**Fig.3**) and ATAC-seq data (**Fig.5**) already provides strong validation of most data in **Fig.6**, given that ATAC-seq peaks almost always co-localize with H3K27ac peaks. We also investigated the possibility of performing KMT2D-CREBBP Co-IPs in mouse GC B-cells, but unfortunately the antibodies do not work as robustly against murine proteins.

R1.11. Have the authors acquired data for chronic GCs as detectable in mesLN (or PP)?

Answer: We thank the reviewer for this question. We don't know if there were chronic GCs since we don't have tissues available for examination.

R1.12. Is class switch recombination undisturbed upon Crebbp and Kmt2d haploinsufficiency? Impaired IgG1 switching might impact on the results in Suppl Fig 3 e-j.

Answer: We thank the reviewer for this question. We didn't see class switching defect upon CK haploinsufficiency. On the contrary, there were more IgG1-switched GCB cells in CK than WT (**Revision Fig. 9**). Class switching defect was only observed with homozygous KMT2D deletion, as we reported previously⁶, but it is evident this does not happen in the heterozygous state reported here. We have included these data in the revised manuscript as **Supplementary Fig. 4k-m**.

Revision Fig. 9. a, Representative FACS plots show the gating strategy and frequency of splenic IgM+ and IgG1+ GCB cells in mice at day 10 post SRBC. **b-c**, FACS analysis showing the relative abundance of GCB (**b**, normalized to B220+), IgM+ and IgG1+ GCB (**c**, normalized to GCB) in mice at day 10 post SRBC (mean \pm SD). Each dot represents a mouse (n=5 mice per genotype). Statistical significance was determined by two-tailed unpaired student's t test (*p < 0.05).

Reviewer #2 (Remarks to the Author):

Li and colleagues conducted in vivo experiments to investigate the mechanisms of the concurrent mutations in CREBBP and KMT2D in both GC development and lymphomagenesis. They observed that the loss of CREBBP and KMT2D leads to a more severe lymphoma phenotype and unexpected immune evasion behavior in cancer cells. This loss results in reduced infiltration of CD8+ T cells and suppression of immune synapse genes, contributing to a weakened T-cell response and flawed cell fate decisions, possibly aiding cancer cells in eluding immune surveillance. Additionally, from an epigenetic standpoint, the loss of cooperation between C+K had a significant and profound effect on super enhancers, especially those responsible for regulating immune synapse signaling genes. Functional characterization of the mutations identified in lymphoma derived from humans is essential for a more comprehensive understanding of these genes' roles in lymphomagenesis. **The manuscript is well-crafted**, and some comments are listed below:

Answer: We very much appreciate the reviewer's positive comments!

R2.1. On page 6, lines 3-5, the authors calculated the p-values for the co-occurrence of CREBBP and KMT2D mutations in FL. However, **the text does not specify the number of genes (patients?) that were selected for this calculation**. Moreover, it remains unclear whether the authors took into account the potential functional effects of the mutations, such as **whether they were damaging or not**, in their analysis.

Answer: We thank the reviewer for this question. Significance of co-occurrence was evaluated using Fisher's exact test and only CREBBP and KMT2D mutations were evaluated. FL_all merged dataset in **Supplementary Fig.1a** comprised the following cohorts:

FL_cohort_1, n=26

FL_cohort_2, n=57

Green_FI_PNAS_2015, n=109

Han_BloodCancerDiscovery, n=19

Pasqualucci_CellReports_2014, n=45

Okosun_NatGenet_2014, n=24

Ma_Haematologica_2022, n=198

In total, FL_all contains 478 patients, out of which 250 (52.3%) carry CREBBP mutation, 305 (63.8%) carry KMT2D mutation, 185 (38.7%) carry both mutations. We have included these patient numbers and percentages in the revised **Supplementary Fig.1a** legend.

Given that SIFT or PP2_HDIV_PRED based toxicity prediction is not available for these cohorts, we made no assumptions regarding the deleterious nature of CREBBP and KMT2D mutations when performing co-occurring analysis.

R2.2. In Supplementary Fig. 1d, the authors concluded that BCL2+CK exhibited 'highly, larger, more' features compared to others. However, the conclusion **lacks statistical support**, leaving the basis for this assertion unclear.

Answer: We thank the reviewer for raising this concern. The reported histological analyses were performed by our co-author and internationally recognized hematopathologist Dr. Amy Chadburn, who is a member of the WHO lymphoma classification team⁷. Dr. Chadburn made this statement based on her observation that most BCL2+CK spleens exhibited these features. This is how such diagnostic determinations are made in routine clinical practice. Given that it is technically challenging to quantitatively measure cells exhibiting highly heterogeneous morphology, we have included this caveat statement: “based on detailed hematopathology reporting” in our revised text.

R2.3. On page 7, line 11, the authors noted that the SHM (somatic hypermutation) burden varied among different genotypes. **However, they did not specify which isotype of VH was analyzed.** Understanding whether the VH isotypes were class-switched (e.g., IgG) or not (e.g., IgM) would be relevant, as it could significantly influence the SHM levels.

Answer: We thank the reviewer for pointing this out. Our PCR primers target Ig heavy chain VDJ region, which cannot distinguish different Ig heavy chain isotypes. To address this Ig isotype composition question, we checked the expression levels of different Ig constant genes (encoding different isotypes) from our day 235 murine lymphoma RNA-seq dataset. We found that the Ig isotype composition across all samples was rather heterogeneous, without bias towards any particular isotype selection (**Revision Fig. 10**), suggesting the observed heavier SHM burden is not likely due to changes in Ig class switching. In addition, the link between Ig class switching and SHM is elusive since Ig class switching recombination was shown to frequently occur prior to GC reaction⁸. We have included these data in the revised manuscript as **Supplementary Fig. 1j**.

Revision Fig. 10. Stacked bar plots showing the fraction of different Ig heavy chain isotype genes in day 235 murine lymphoma samples based on RNA-seq data.

R2.4. In the case of human **FL patients**, the authors do not make it clear whether they observed **any differences in prognosis among the C, K, CK, and WT patients**. Likewise, the manuscript **lacks**

information on whether similar conclusions were reached regarding CD8+ T cells in the different mutation groups of FL or DLBCL.

Answer: We appreciate the reviewer's suggestion. Unfortunately, there are still very limited instances of clinically annotated and sequenced FL patients, which makes it hard to address such issues in a statistically robust manner. Moreover, FL treatment is quite heterogeneous, may involve many different and sequential therapeutic regimens, and outcomes are further dependent on how tumors with particular mutational constellations specifically respond to any one of the various therapeutic options. These considerations significantly limit biological conclusions that can be drawn based on currently available data.

However, to gain at least some insight into this question we initiated a new collaboration for this revision with Dr. Oliver Weigert (Ludwig Maximilian University of Munich), who is a leader in the FL genetic biomarkers field. His group conducted an outcomes analysis using their published GLSG2000 trial dataset together with an additional British Columbia Cancer Agency (BCCA) FL dataset⁹. Importantly, most patients in these datasets had advanced FL requiring R-CHOP-like systemic treatment. As such, outcomes are based on more DLBCL-like FL stages which may or may not reflect outcomes as examined from more indolent stages of disease.

Because of these considerations we used failure free survival as an endpoint (very similar to PFS and TTF), since this endpoint is probably clinically most meaningful in this context. OS is heavily biased by subsequent therapies (especially in indolent diseases like FL). Also, given the very high response rates to R-CHOP-like regimens, ORR etc. is very unlikely to show differences and is clinically not relevant. This analysis revealed that patients carrying C, K, or CK mutations showed equivalent trends to their WT counterparts (**Revision Fig. 11**).

There are several caveats in this analysis that could explain the minor differences observed. 1) The patient numbers are highly unbalanced among different groups due to the high frequency of C/K mutations in FL. 2) Disease in WT patients must be driven by other mutations, with distinct biological dependencies and functions, which will affect response to specific treatments. 3) There is insufficient statistical power to consider multiple covariates such as mutation burden, gender, age, health history etc. among the different groups.

Finally, CD8 staining was not available for these cohorts and so could not be factored in, again even where they available such analyses would be affected by the same limitations mentioned above. Given all of this we have not included these findings in the revised manuscript, although Dr. Weigert's group is acknowledged for the time and effort they devoted to addressing this question.

Revision Fig. 11. Failure free survival curve of different FL patients, stratified into 4 groups based on their C/K mutation status, in GLSG2000 cohort (a) and BCCA cohort (b). p values were calculated by log-rank test.

R2.5. The authors concluded that the loss of function in CREBBP and KMT2D cooperates to accelerate FL development, manifesting more aggressive characteristics than either allele alone. **However, Figure 1b and 1d do not visibly differentiate between BCL2+k and BCL2+CK. An explanation from the authors may be needed to clarify this apparent inconsistency.**

Answer: We understand the reviewer's concern. We agree that the difference between BCL2+K and BCL2+CK in Fig.1b and 1d is minor, yet the trend that BCL2+CK is more deleterious than BCL2+K is clearly visible. Furthermore, we provide many other data (including new results from this revision) showing the more aggressive phenotype in BCL2+CK, such as histology, SHM, survival, reduced clonality, and CD8 reduction and exhaustion (Fig. 1e, 1f, 1h-n, Supplementary Fig. 2a-b).

R2.6. It is unclear from the manuscript whether the authors **examined the phenotype of CD8 or CD4 T cells in the BCL2+CK mice** or human FLs, such as **signs of exhaustion or hyperactivity**

Answer: We thank the reviewer for this great suggestion. As mentioned for reviewer 1 (R1.7): We performed Cytek multi-dimensional flow cytometry analysis of T-cell phenotypes (naive, central memory, effector, exhaustion, regulatory T cells, Tfh, Tfr) using the remaining frozen splenocyte specimens from our lymphoma cohort mice at both time points to gain more insights into how CK loss of function differentially affects CD4 and CD8 T cells in the lymphoma microenvironment. As noted before, CD8, but not CD4, displayed progressive reduction from BCL2+C, BCL2+K, to BCL2+CK, relative to BCL2 at both time points (Revision Fig. 4a-b). Given that all subsequent analyses were based on these specimens, we replaced the former flow cytometry analysis from Fig. 1g-h with these data, to directly match the more detailed flow cytometry results.

Revision Fig. 4. FACS analysis showing the relative abundance of splenic CD4+ (a) or CD8+ (b) cells normalized to CD45+ cells (mean \pm SD) in mice from VavPBCL2 lymphoma cohort. Each dot represents a mouse (n=4 mice per genotype). Statistical significance was determined using ordinary one-way ANOVA followed by Tukey-Kramer's multiple comparisons test (*p < 0.05; ***p < 0.001; ****p < 0.0001).

On the CD8 side, at day 116 we observed significant expansion of CD8 effector cells (CD8+CD44+CCR7-), accompanied by a corresponding reduction of central memory (CM, CD8+CD44+CCR7+) CD8 cells in BCL2+CK, that was greater than effects observed in BCL2+C and BCL2+K (**Revision Fig. 5a-b**).

We note for the reviewer that day 235 specimens were actually thawed and examined in our lab prior to the day 116 tumors mentioned just above. Unfortunately, we were unable to accurately separate naive, CM and effector cells at day 235 due to failure of the CD62L channel, caused by the known pitfall of shedding and loss of this marker (that we were initially unaware of) after freeze-thaw cycle as previously reported². This was why we switched to subsequent labeling of CCR7 in our day 116 samples.

This issue notwithstanding, at day 235 we noted significant expansion of CD44+ activated CD8 T cells (which contain both effector and CM populations) in BCL2+C and BCL2+CK, whereas the fraction of these cells in BCL2+K was reduced compared to BCL2 (**Revision Fig. 5c-d**). This interpretation is limited by the lack of CD62L staining at day 235 and so should not be viewed as conclusive.

Most remarkably, even though the abundance of CD8 cells was strongly reduced at day 235 in the BCL2+CK setting, **there was a massive and significant increase in the proportion of these cells manifesting an exhausted phenotype**, whereas the proportion of exhausted cells was roughly similar to BCL2 alone in the BCL2+C and BCL2+K setting (**Revision Fig. 5e-f**). In contrast there was no change in the abundance of exhausted CD8 cells at day 116. *Although we do not know the etiology of this phenotypic discovery, it does indicate significant CD8 T-cell dysfunction in the BCL2+CK setting. Lacking more specific information on the fraction of effector vs CM cells in this context we suggest it is best to avoid concluding at this point that the cause of exhaustion is linked to the apparent increase in activated CD8 T-cells. However, it does further underline how cooperative effects of C+K can alter the immune landscape and we thank the reviewer for encouraging us to add these interesting experiments to the manuscript.*

Revision Fig. 5. FACS-based immunophenotyping of splenic CD8 T cells from murine VavPBCL2 lymphoma cohort. **a, c, e**, Representative FACS plots show the gating strategy and frequency of different splenic CD8 subtypes (**a**, naïve/CM/effector CD8 at day 116, **c**, CD44+ activated CD8 at day 235, **e**, TCF1-TOX+ exhausted CD8 at day 116 and 235) in indicated mice. **b, d, f**, FACS analysis showing the relative abundance of splenic naïve/CM/effector CD8 at day 116 (**b**), CD44+ CD8 at day 235 (**d**), or TCF1-TOX+ exhausted CD8 at day 116 and 235 (**f**) normalized to CD8+ cells (mean \pm SD). Each dot represents a mouse (n=4 mice per genotype). Statistical significance in panels **b, d, f** was determined using ordinary one-way ANOVA followed by Tukey-Kramer's multiple comparisons test (*p < 0.05; **p < 0.01; ***p < 0.001; ****p < 0.0001).

On the CD4 side, TFH (T follicular helper, CD4+CXCR5+PD1+FOXP3-) cell frequency was progressively increased from BCL2+C, BCL2+K, to BCL2+CK, relative to BCL2 at both time points (**Revision Fig. 6a-c**). Intriguingly, TFH expansion occurred out of proportion with GC suppressive T follicular regulatory cells (TFRs, CD4+CXCR5+PD1+FOXP3+) at day 116 (**Revision Fig. 6d-e**). Given that TFRs are known to terminate the GC reaction in part by suppressing TFH functions^{3, 4}, we speculate that increased TFH/TFR ratios may favor initial expansion of malignant GC-like structures. Consistently, TFH/GCB ratios were significantly increased in BCL2+K and BCL2+CK compared to BCL2 at day 116 (**Revision Fig. 6f**). Finally, there was an equivalent increase in

Tregs (CD4+FOXP3+) among C/K/CK mutant genotypes relative to BCL2 in the end-stage lymphomas (**Revision Fig. 6g-h**), pointing to additional layers of immune dysfunction.

Revision Fig. 6. FACS-based immunophenotyping of splenic CD4 T cells from murine VavPBCL2 lymphoma cohort. **a-b**, Representative FACS plots show the gating strategy and frequency of splenic Tfh (CD4+CXCR5+PD1+FOXP3-) and Tfr (CD4+CXCR5+PD1+FOXP3+) cells in indicated mice at day 116 (**a**) or day 235 (**b**) post BMT. **c-d**, FACS analysis showing the relative abundance of splenic Tfh (**c**) or Tfr (**d**) normalized to CD4+ cells at day 116 and 235 post BMT (mean \pm SD). Each dot represents a mouse (n=4 mice per genotype). **e-f**, FACS analysis showing the ratio of Tfh vs Tfr (**e**) or Tfh vs GCB (**f**). **g**, Representative FACS plots show the gating strategy and frequency of splenic Treg (CD4+FOXP3+) in indicated mice at day 116 and 235 post BMT. **h**, FACS analysis showing the relative abundance of splenic Treg normalized to CD4+ cells at day 116 and 235 post BMT (mean \pm SD). Statistical significance in panels **c-f** and **h** was determined using ordinary one-way ANOVA followed by Tukey-Kramer's multiple comparisons test (*p < 0.05; **p < 0.01; ***p < 0.001; ****p < 0.0001).

Together, these data suggest that CREBBP and KMT2D loss of function cooperatively remodels the immune microenvironment from a TFH-enriched pro-GC reaction early stage to a CD8-exhausted and depleted later stage to facilitate lymphomagenesis (**Revision Fig. 7**). These new CD4 and CD8 flow data are included in the revised manuscript as **Fig. 1g-j, 1o, Supplementary Fig. 2c-o**.

Lymphoma microenvironment change in BCL2+CK vs BCL2

Revision Fig. 7. A graphical model summarizing the lymphoma microenvironment change in BCL2+CK vs BCL2. Up and down arrows indicate population increase or decrease respectively in BCL2+CK vs BCL2. CD8cm: central memory CD8 (CD8+CD44+CCR7+), CD8eff: effector CD8 (CD8+CD44+CCR7-), CD8act: activated CD8 (CD8+CD44+, including both CM and effector populations), CD8ex: exhausted CD8 (CD8+TCF1-TOX+).

R2.7. On page 7, the flow data reveals a similar abundance of CD4+ T cells but a reduction in CD8+ T cells in BCL2+CK mice. It would be crucial for the authors to specify which CD8+ cell subtypes were reduced, as this information could enhance the understanding of the immune response dynamics.

Answer: We thank the reviewer for this suggestion. As mentioned above we observed an elevated fraction of effector CD8 and a concordantly reduced fraction of CM CD8 with progressively increasing magnitude from BCL2+C, BCL2+K, to BCL2+CK relative to BCL2 at day 116 (**Revision Fig. 5a-b**).

However, to gain a sense of overall abundance of each CD8 T cell subtype, we also normalized them to total lymphocytes (CD45+) using our additional specimens described above. At day 116, all CD8 subtypes including exhausted cells displayed a progressive reduction pattern across the genotypes reflecting their respective total CD8 abundance (**Revision Fig. 12a-c**). As expected, at day 235, the frequency of both activated (CD44+) and exhausted CD8 were reduced in BCL2+CK, albeit to different degrees (**Revision Fig. 12d-f**).

Revision Fig. 12. FACS-based immunophenotyping of splenic CD8 T cells from murine VavPBCL2 lymphoma cohort. **a, d**, FACS analysis showing the relative abundance of splenic CD8+ cells normalized to CD45+ cells at day 116 (**a**) or 235 (**d**) (mean \pm SD). **b**, FACS analysis showing the relative abundance of different CD8 subtypes (Naive: CCR7+CD44-; CM: CCR7+CD44+; Effector: CCR7-CD44+) normalized to CD45+ cells at day 116 (mean \pm SD). **c, f**, FACS analysis showing the relative abundance of splenic TCF1-TOX+CD8+ exhausted cells normalized to CD45+ cells at day 116 (**c**) or 235 (**f**) (mean \pm SD). **e**, FACS analysis showing the relative abundance of CD44+CD8+ cells normalized to CD45+ cells at day 235 (mean \pm SD). Each dot represents a mouse (n=4 mice per genotype). Statistical significance was determined using ordinary one-way ANOVA followed by Tukey-Kramer's multiple comparisons test (* $p < 0.05$; ** $p < 0.01$; *** $p < 0.001$; **** $p < 0.0001$).

R2.8. On page 16, line 2, the authors performed RNA sequencing on B220+ cells, identifying them as lymphoma cells. However, the manuscript does not specify the criteria used to define these cells as tumor cells.

Answer: We thank the reviewer for pointing this out. B cell lymphomas are highly heterogeneous in terms of maturation/differentiation state, generally encompassing a mixture of B-cell states including CB, CC, memory, and plasmablasts. As such, we included all B220+ cells for RNA-seq analysis. Note that the massive tumor burden and tissue disruption observed in these animals makes it exceedingly unlikely that there is significant signal from residual normal mature B-cells.

R2.9. On page 18, the authors mentioned impaired expression of key genes involved in GC B cell interaction with TFH cells and GC exit. However, how C+K differentially modulate the infiltration

of CD8+ and CD4+ T cells remains unclear, given that BCL2+CK mice mainly had reduced CD8+ T cells rather than CD4+ T cells. Further clarification on this aspect is needed.

Answer: We thank the reviewer for raising this point. Although we don't know exactly how this differential effect on CD4 vs CD8 T cells is achieved at the molecular level, we speculate that at the cellular level, this is because CK GCB/lymphoma cells are highly dependent on TFH cells for expansion as supported by 1) hypersensitivity to CD40/CD40L blockade (**Fig. 4h-j**), and 2) aberrant TFH expansion relative to BCL2 alone (**Revision Fig. 6a-f**). As such, reduction in CD4 abundance would not be expected to favor transformation of CK deficient GC B-cells.

In contrast, CD8 cells are not normally part of the GC reaction and are more likely to exert an anti-tumor cytotoxic effect. Our data suggest that as they evolve, CK lymphomas may employ several integrated strategies to reduce CD8 abundance and cytotoxic attack, including Treg expansion (**Revision Fig. 6g-h**), chronic suboptimal CD8 stimulation (**Revision Fig. 8**) and enhanced CD8 exhaustion (**Revision Fig. 5e-f**).

Together, we hypothesize that the impaired expression of immune synapse genes we report would somehow be tuned to reduce immune tone and hence block CD8 recruitment without driving away the TFH cells. Since there are many such genes involved and the combinatorial complexity is quite daunting, we are not able to explain which subsets of these genes explain the phenotype. These considerations are further explored in the **revised discussion section**.

Reviewer #3 (Remarks to the Author):

In this interesting manuscript, Li and colleagues investigate the effects of combined heterozygous loss of CREBBP and KMT2D in lymphoma, using the VavP-Bcl2 model and the GC B-cell specific Cg1-Cre strain. Using a variety of in vitro and in vivo analyses, authors show that combined loss of CREBBP and KMT2D induces a more severe phenotype than loss of each of them separately. This is associated to a phenotype of immune evasion with drastically reduced infiltrating CD8 T-cells. Furthermore, authors show the cooperative nature of CREBBP and KMT2D in the regulation of their epigenetic target programs, and even demonstrate that CREBBP and KMT2D co-interact. Finally, the epigenetic phenotype upon combined KMT2D/CREBBP loss was especially strong at superenhancer regions (and, more specifically, at super enhancers driving the expression of immune synapse signaling genes). **Overall, this is a very comprehensive study uncovering novel relevant biology, and I would like to congratulate authors for their really nice work.**

Answer: We appreciate the reviewer's overall enthusiasm and positive comments on our work.

R3.1. My only minor comment to authors is that it would be good if they could **show all 4 genotypes in Fig 4i** (as of know, it is not clear if the anti-CD40 bar is grouping all 4 genotypes, or if it's just one genotype only).

Answer: We thank the reviewer for this question. The anti-CD40 bar in old Fig. 4i only contains WT+CK BM chimeric mice. To address this point, we repeated mixed chimera BMT and CD40/CD40L blocking experiments (using an anti-CD40L blocking antibody to block CD40/CD40L interactions) adding the C vs WT and K vs WT comparisons. Similar to our previous CK vs WT fitness study, we mixed C or K BM with WT BM expressing distinct congenic markers at equal ratios, then injected them into lethally irradiated syngeneic recipient mice. Fully engrafted recipient mice were immunized with SRBC. Anti-CD40L blocking antibody or IgG control were then administered at days 4 and 6 post-immunization (**Revision Fig. 13a**).

In IgG-treated mice, CK deficient GCB cells exhibited a significant fitness advantage over their WT controls. K deficient GCB cells also gained fitness advantage although to a lesser extent than CK, whereas C deficiency alone failed to confer such advantage (**Revision Fig. 13b-d**), consistent with the stepwise GCB expansion from C, K to CK relative to WT GEMM (**Fig. 2d**).

Upon CD40L blockade, we observed significantly reduced advantage among the CK GCB cells, whereas there was only a mild trend towards advantage reduction vs WT among the K GCB cells (**Revision Fig. 13c-d**). In contrast, there was a mild but significant fitness disadvantage in the C only GCB cells. This suggests that the increased CD40L blockade sensitivity observed in CK may derive more from the C component although K probably also contributes. These intriguing results are added to the revised manuscript (**Fig. 4h-j, Supplementary Fig. 4n**).

Revision Fig. 13. Fitness study in the absence or presence of CD40/CD40L blockade using BM chimeric mice. **a**, Experimental scheme for CD40/CD40L blocking assay. **b**, FACS data showing the frequency of GCB cells among total B cells in either control IgG or CD40L blocking antibody treated mice, two-tailed unpaired t test. **c**, FACS data showing the ratio of WT (black dots), C (blue dots), K (orange dots) and CK (red dots)-derived GCB cell percentage to their respective parental total B cell percentage in either control IgG or CD40L blocking antibody treated mice. Each pair of connected dots represents a mouse, two-tailed paired t test. **d**, GCB (CD45%) / B cell (CD45%) ratio fold change (C/WT, K/WT, and CK/WT) in either control IgG or CD40L blocking antibody treated mice, two-tailed unpaired t test. (**p < 0.01; ***p < 0.001; ****p < 0.0001).

R3.2. Similarly, it would be good if authors could show the other 2 genotypes in Fig 4j, beyond the WT and CK, so that we can evaluate potential differences between single and combined genotypes.

Answer: We thank the reviewer for this suggestion, which was addressed just above.

References

1. Vlasevska S, *et al.* KMT2D acetylation by CREBBP reveals a cooperative functional interaction at enhancers in normal and malignant germinal center B cells. *Proc Natl Acad Sci U S A* **120**, e2218330120 (2023).
2. Florek M, *et al.* Freeze and Thaw of CD4+CD25+Foxp3+ Regulatory T Cells Results in Loss of CD62L Expression and a Reduced Capacity to Protect against Graft-versus-Host Disease. *PLoS One* **10**, e0145763 (2015).
3. Jacobsen JT, *et al.* Expression of Foxp3 by T follicular helper cells in end-stage germinal centers. *Science* **373**, (2021).
4. Linterman MA, *et al.* Foxp3+ follicular regulatory T cells control the germinal center response. *Nat Med* **17**, 975-982 (2011).
5. Hoekstra ME, Vijver SV, Schumacher TN. Modulation of the tumor micro-environment by CD8(+) T cell-derived cytokines. *Curr Opin Immunol* **69**, 65-71 (2021).
6. Ortega-Molina A, *et al.* The histone lysine methyltransferase KMT2D sustains a gene expression program that represses B cell lymphoma development. *Nat Med* **21**, 1199-1208 (2015).
7. Alaggio R, *et al.* The 5th edition of the World Health Organization Classification of Haematolymphoid Tumours: Lymphoid Neoplasms. *Leukemia* **36**, 1720-1748 (2022).
8. Roco JA, *et al.* Class-Switch Recombination Occurs Infrequently in Germinal Centers. *Immunity* **51**, 337-350 e337 (2019).
9. Pastore A, *et al.* Integration of gene mutations in risk prognostication for patients receiving first-line immunochemotherapy for follicular lymphoma: a retrospective analysis of a prospective clinical trial and validation in a population-based registry. *Lancet Oncol* **16**, 1111-1122 (2015).

REVIEWERS' COMMENTS

Reviewer #1 (Remarks to the Author):

I would like to thank and congratulate the authors for the revised manuscript.

The manuscript improved significantly by the novel experimental data, especially the characterization of the lymphoma micro-environment.

Although recurrent mutations were not identified in the WES analysis of mouse tumors, I would like to encourage the authors to make the sequencing data available for the scientific community and include it in the revised manuscript.

Besides this suggestion I have no additional comments.

Reviewer #3 (Remarks to the Author):

Authors have addressed all my comments and other reviewers' comments. Congratulations again.

Reviewer #1 (Remarks to the Author):

I would like to thank and congratulate the authors for the revised manuscript. The manuscript improved significantly by the novel experimental data, especially the characterization of the lymphoma micro-environment. Although recurrent mutations were not identified in the WES analysis of mouse tumors, I would like to encourage the authors to make the sequencing data available for the scientific community and include it in the revised manuscript. Besides this suggestion I have no additional comments.

Answer: We appreciate the reviewer's positive comments on our work. We included the mouse tumor mutation data as a Supplementary Data and deposited the raw WES data to GEO database.

Reviewer #3 (Remarks to the Author):

Authors have addressed all my comments and other reviewers' comments. Congratulations again.

Answer: We thank the reviewer's positive comments on our work.